# USING CONTRASTIVE LEARNING WITH GENERATIVE SIMILARITY TO LEARN SPACES THAT CAPTURE HUMAN INDUCTIVE BIASES

## ABSTRACT

Humans rely on strong inductive biases to learn from few examples and abstract useful information from sensory data. Instilling such biases in machine learning models has been shown to improve their performance on various benchmarks including few-shot learning, robustness, and alignment. However, finding effective training procedures to achieve that goal can be challenging as psychologically-rich training data such as human similarity judgments are expensive to scale, and Bayesian models of human inductive biases are often intractable for complex, realistic domains. Here, we address this challenge by introducing a Bayesian notion of generative similarity whereby two datapoints are considered similar if they are likely to have been sampled from the same distribution. This measure can be applied to complex generative processes, including probabilistic programs. We show that generative similarity can be used to define a contrastive learning objective even when its exact form is intractable, enabling learning of spatial embeddings that express specific inductive biases. We demonstrate the utility of our approach by showing that it can be used to capture human inductive biases for geometric shapes, distinguish different abstract drawing styles that are parameterized by probabilistic programs, and capture abstract high-level categories that enable generalization.

## 1 INTRODUCTION

Human intelligence is characterized by strong inductive biases that enable humans to form meaningful generalizations (Tenenbaum et al., 2011; Lake et al., 2015), learn from few examples (Lake et al., 2015), and abstract useful information from sensory data (Gershman, 2017). Instilling such biases into machine learning models has been at the center of numerous recent studies (Kumar et al., 2022; McCoy et al., 2020; Peterson et al., 2019; Hebart et al., 2020; Sucholutsky et al., 2023a; Sucholutsky & Griffiths, 2024; McCoy & Griffiths, 2023; Muttenthaler et al., 2024; Snell et al., 2023; Jha et al., 2023), and has been shown to improve accuracy, few-shot learning, interpretability, and robustness (Sucholutsky et al., 2023b). Key to this effort is the ability to find effective training procedures to imbue neural networks with these inductive biases. Two prominent approaches for this are i) leveraging the literature on modeling human inductive biases with Bayesian models (Tenenbaum et al., 2011; Griffiths et al., 2010) to specify a computational model for the bias of interest and then distilling it into the model, usually via meta-learning (Kumar et al., 2022; Binz et al., 2023; McCoy et al., 2020), and ii) incorporating psychologically-rich human judgments in the training objectives of models such as soft labels (Sucholutsky & Schonlau, 2021), categorization uncertainty (Collins et al., 2023; Peterson et al., 2019), language descriptions (Kumar et al., 2022; Marjieh et al., 2023), and similarity judgments (Muttenthaler et al., 2024; Jha et al., 2023; Esling et al., 2018).

While both approaches are promising, they are not without limitations. Though Bayesian models provide an effective description of human inductive biases, they are often computationally intractable due to expensive Bayesian posterior computations that require summing over large hypothesis spaces. This problem is particularly pronounced when considering symbolic models which are often used to describe human inductive biases (Lake et al., 2019; 2017; Sablé-Meyer et al., 2021; 2022; Quilty-Dunn et al., 2023). Likewise, incorporating human judgments in model objectives may be intuitive, but it is often not scalable for the data needs of modern machine learning. For example, while incorporating human similarity judgments (i.e., judgments of how similar pairs of stimuli are) has

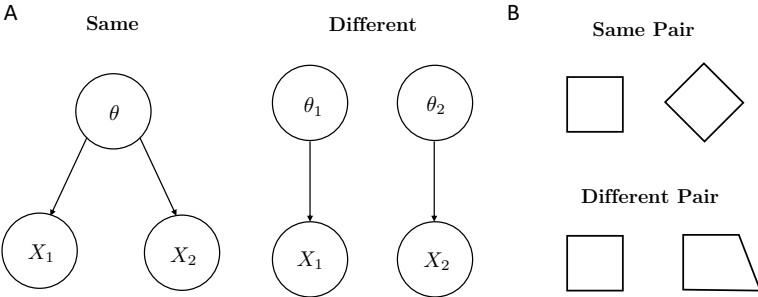

Figure 1: **Schematic representation of generative similarity. A.** Graphical models for the same and different data generation hypotheses. **B.** Example same and different quadrilateral shape pairs.

been shown to improve model behavior (Esling et al., 2018; Jha et al., 2023; Muttenthaler et al., 2024), collecting them at the scale of modern datasets is challenging as the number of required judgments grows quadratically in the number of stimuli (though see Marjieh et al. (2023) for proxies).

Here, we introduce a third approach for instilling inductive biases based on the method of contrastive learning (Chen et al., 2020), a widely used training procedure in machine learning. Contrastive learning uses the designation of datapoints as being the "same" or "different" to learn a representation of those datapoints where the "same" datapoints are encouraged to be closer together and "different" datapoints further apart. This approach provides a way to go from a similarity measure to a representation. We define a principled notion of similarity based on Bayesian inference (initially proposed in Kemp et al., 2005) and show that it can be naturally implemented in a contrastive learning framework even when its exact form is intractable. Specifically, given a set of samples and a hierarchical generative model of the data from which data distributions are first sampled and then individual samples are drawn (e.g., a Gaussian mixture), we define the *generative similarity* between a pair of samples to be their probability of having been sampled from the same distribution relative to that of them being sampled from two independently-drawn distributions (Figure 1). By using Bayesian models to define similarity within a contrastive learning framework, we provide a general procedure for instilling human inductive biases in machines.

To demonstrate the utility of our approach, we apply it to four domains of increasing complexity. First, we consider a Gaussian mixture example where similarity and embeddings are analytically tractable, which we then further test with simulations. Second, we consider a generative model for quadrilateral shapes where generative similarity can be computed in closed form and can be incorporated explicitly in a contrastive objective. By training a model with this objective, we show that it acquires human-like regularity biases in a geometric reasoning task. Third, we consider probabilistic programs, using two classes of probabilistic programs from DreamCoder (Ellis et al., 2021; Sablé-Meyer et al., 2022). While generative similarity is not tractable in this case, we show that it can be implicitly induced using a Monte Carlo approximation applied to a triplet loss function, leading to a representation that better captures the structure of the programs compared to standard contrastive learning. Fourth, we show the scalability of this approach by applying our method to a standard large-scale machine learning dataset, ImageNet (Deng et al., 2009), and demonstrate how generative similarity can allow for the learning of high-level, hierarchical image categories that enable generalization. Viewed together, these results highlight a path towards alignment of human and machine intelligence by instilling useful inductive biases from Bayesian models of cognition into machine models via a scalable contrastive learning framework, and allowing neural networks to capture abstract domains that previously were restricted to symbolic models.

## 2 GENERATIVE SIMILARITY AND CONTRASTIVE LEARNING

We begin by laying out the formulation of generative similarity and its integration within a contrastive learning framework. Given a set of samples $D$ and an associated generative model of the data $p(D) = \int p(D|\theta)p(\theta)d\theta$ where $p(\theta)$ is some prior over distribution parameters (e.g., a beta prior) and $p(D|\theta)$ is an associated likelihood function (e.g., a Bernoulli distribution), we define the *generative*

*similarity* between a pair of samples $s_{\text{gen}}(x_1, x_2)$, to be the Bayesian probability odds ratio for the probability that they were sampled from the same distribution to that of them being sampled from two independent (or "different") distributions

$$s_{\text{gen}}(x_1, x_2) = \frac{p(\text{same}|x_1, x_2)}{p(\text{different}|x_1, x_2)} = \frac{\int p(x_1|\theta)p(x_2|\theta)p(\theta)d\theta}{\int p(x_1|\theta_1)p(x_2|\theta_2)p(\theta_1)p(\theta_2)d\theta_1 d\theta_2} \tag{1}$$

where we assume that *a priori* $p(\text{same}) = p(\text{different})$ (i.e., the prior over the two hypotheses is uniform). The same and different data generation hypotheses are shown in Figure 1A along with example same and different pairs in Figure 1B in the case of a generative process of quadrilateral shapes, with the same pair corresponding to two squares, and the different pair corresponding to a square and a trapezoid. This definition builds on existing Bayesian models of similarity in cognitive science (Shepard, 1987; Tenenbaum & Griffiths, 2001), and in particular Kemp et al. (2005).

Given the definition of generative similarity, we next distinguish between two scenarios. If $s_{\text{gen}}$ is tractable, then its incorporation in a contrastive loss function is straightforward: given a parametric neural encoder $\phi_\varphi(x)$ and a prescription for deriving similarities from these embeddings, e.g. $s_{\text{emb}} = s_0 e^{-d}$ where $d$ is a distance measure and $s_0$ is a constant, we can then directly optimize the embedding parameters such that the difference between the generative similarity and the corresponding embedding similarity is minimized, e.g.,

$$\varphi^* = \arg\min_\varphi \mathbb{E}_p(s_{\text{emb}}(\phi_\varphi(X_1), \phi_\varphi(X_2)) - s_{\text{gen}}(X_1, X_2))^2. \tag{2}$$

If, on the other hand, $s_{\text{gen}}$ is not tractable, we can implicitly incorporate it in a neural network using individual triplet loss functions (here we focus on triplets for convenience, but our formalism can be easily adapted to larger sample tuples). Specifically, given a generative model of the data, we can define a corresponding *contrastive* generative model on data triplets $(X, X^+, X^-)$ as follows

$$p_c(x, x^+, x^-) = \int p(x|\theta^+)p(x^+|\theta^+)p(x^-|\theta^-)p(\theta^+)p(\theta^-)d\theta^+ d\theta^- \tag{3}$$

and then given a choice of a triplet contrast function $\ell$, e.g., $d(\phi(x), \phi(x^+)) - d(\phi(x), \phi(x^-))$ or some monotonic function of it (alternatively, one could also use an embedding similarity measure such as the dot product; Sohn, 2016), we define the optimal embedding to be

$$\varphi^* = \arg\min_\varphi \mathbb{E}_{p_c}\ell(\phi_\varphi(X), \phi_\varphi(X^+), \phi_\varphi(X^-)) \tag{4}$$

Crucially, this function can easily be estimated via Monte Carlo with triplets sampled from

$$\begin{cases} \theta^+, \theta^- \sim p(\theta), & \text{sample distributions} \\ x, x^+ \sim p(x|\theta^+), & \text{sample 'same' examples} \\ x^- \sim p(x|\theta^-), & \text{sample 'different' examples} \end{cases} \tag{5}$$

The functional in Equation 4 has a desirable property: in Appendix A we prove that if $\ell$ is chosen to be convex and strictly increasing in $\Delta_\phi(x, x^+, x^-) \equiv d(\phi(x), \phi(x^+)) - d(\phi(x), \phi(x^-))$ (e.g., softmax loss $\ell(\Delta_\phi) = \log(1 + \exp(\Delta_\phi))$; Sohn, 2016), then the optimal embedding that minimizes Equation 4 ensures that the expected distance between same pairs is strictly smaller than that of different pairs as defined by the processes in Figure 1A, i.e.,

$$\mathbb{E}_{p_{\text{same}}}d(\phi^*(X), \phi^*(X^+)) < \mathbb{E}_{p_{\text{diff}}}d(\phi^*(X), \phi^*(X^-)). \tag{6}$$

## 2.1 RELATED WORK

We next discuss how our framework for instilling human inductive biases connects to existing contrastive techniques. First, observe that by carrying out the integration in Equation 1 we have $s_{\text{gen}}(x_1, x_2) = p(x_1, x_2)/p(x_1)p(x_2)$ where $p(x_1, x_2) = \int p(x_1|\theta)p(x_2|\theta)p(\theta)d\theta$ and $p(x_{1,2})$ are its marginals. This means that (up to a logarithm) generative similarity is a specific type of point-wise mutual information (Church & Hanks, 1990). While point-wise mutual information (MI) is broadly concerned with sample independence, generative similarity goes further by tying this independence to a hierarchical generative structure that encapsulates an inductive bias associated with the data. This is in line with work suggesting that the success of MI-based contrastive learning approaches hinges

significantly on the specific inductive biases incorporated in the process (Tschannen et al., 2019). Notable contrastive methods with losses based on MI maximization include SimCLR (Chen et al., 2020) and InfoNCE (Oord et al., 2018) more broadly. SimCLR is a training framework whereby a representation is learned by applying various augmentations to the training data and then using those to construct positive and negative pairs that are then incorporated in a contrastive (e.g. triplet) loss. Crucially, positive and negative pairs in SimCLR (and InfoNCE) are usually constructed in a domain-agnostic manner by applying augmentations such as rotations and rescalings to the data. While this is a good default strategy, when there is prior expectation about the structure of the underlying generative process it glosses over finer within- and across-category relations that are explicitly incorporated in our generative similarity sampling scheme (Equation 5). Put differently, SimCLR corresponds to a default generative process in which a base image is sampled and then a random augmentation is applied, as opposed to richer generative models that group the base images themselves into meaningful hierarchies reflecting human inductive biases.

Second, it is possible to show under suitable regularization (to ensure that $s_{\text{gen}}$ can be treated as a distribution) that generative similarity (Equation 1) can be derived as the minimizer of the functional $D_{KL}[s||p_{\text{same}}] - D_{KL}[s||p_{\text{diff}}]$ (see Appendices B-D) which is reminiscent of contrastive divergence learning, a method based on a contrastive objective between two Kullback-Leibler (KL) divergences (Carreira-Perpinan & Hinton, 2005). However, while the above objective is also a contrastive difference between divergences, it is different from that found in CDL in which the goal is to reduce the difference between the divergences whereas in our case it is to increase their (negative) contrast.

Third, our framework is by no means the only contrastive framework that incorporates a Bayesian component. Here it is worth noting two lines of work: (i) Contrastive kernel methods such as the mutual information kernel (MIK) (Seeger, 2001) and the positive pair kernel (PPK) (Johnson et al., 2022). Both MIK and PPK involve a kernel object that is analogous to Equation 1 as the basis of their approach. However, they both differ from our current framework, the former being largely a support vector machine, and the latter considering generative processes in the limited sense of domain-agnostic augmentations (as with SimCLR) and without explicitly incorporating the kernel in a contrastive loss (as with Equations 2-5). It is also unclear from these works why such contrastive kernels can be an effective way of instilling human inductive biases in machines. (ii) Other hybrid Bayesian contrastive learning methods (e.g., Liu & Wang (2023)) devise Bayesian computations to mine hard negative examples to improve contrastive training. However, this approach is quite different from the way Bayesian computations are incorporated in our framework.

## 3 EXPERIMENTS

### 3.1 ILLUSTRATIVE EXAMPLE: GENERATIVE SIMILARITY OF A GAUSSIAN MIXTURE

To get a sense of the contrastive training procedure with generative similarity, we begin with a Gaussian mixture example. Gaussian mixtures are an ideal starting point because i) they are analytically and numerically tractable, and ii) they play a key role in the cognitive literature on models of categorization (Rosseel, 2002; Sanborn et al., 2010). Consider a data generative process that is given by a mixture of two Gaussians with means $\mu_{1,2}$, equal variances $\sigma^2$, and a uniform prior $p_{1,2} = 1/2$. Without loss of generality, we can choose a coordinate system in which $\mu_1 = -\mu_2 \equiv \mu$. Consider further a subfamily of embeddings that are specified by linear projections $\phi_\varphi(x) = \varphi \cdot x$ where $\varphi$ is a unit vector of choice $\varphi \cdot \varphi = 1$. A natural measure of contrastive loss in this case would be

$$\mathbb{E}_{p_c}\ell = \mathbb{E}_{p_c}\left[\phi_\varphi(X) \cdot \phi_\varphi(X^-) - \phi_\varphi(X) \cdot \phi_\varphi(X^+)\right] \tag{7}$$

i.e., using the 'dot' product as a measure of embedding similarity (Sohn, 2016) (for one-dimensional embeddings this is just a regular product). By plugging in the definition of the Gaussian mixture and simplifying using Gaussian moments (see Appendix E), Equation 7 boils down to $\mathbb{E}_{p_c}\ell \propto -4||\mu||_2^2 \cos^2 \theta_{\varphi\mu}$ where $\theta_{\varphi\mu}$ is the angle between $\varphi$ and $\mu$. This loss is minimized for $\cos \theta_{\varphi\mu} = \pm 1$ or equivalently $\varphi^* = \pm\hat{\mu}$ where $\hat{\mu}$ is the normalized version of $\mu$. This means that the optimal linear mapping is simply one that projects different points onto the axis connecting the centers of the two Gaussians $\mu_1 - \mu_2 = 2\mu$, which is equivalent to a linear decision boundary that is orthogonal to the line $\mu_1 - \mu_2$ and passing through the origin, and thus effectively recovering a linear classifier (Figure 2A). We further confirmed in a simulation that this behavior persists when applying the Monte-Carlo approximation in Equation 5 with a quadratic loss to a two-layer perceptron (Figure 2B;

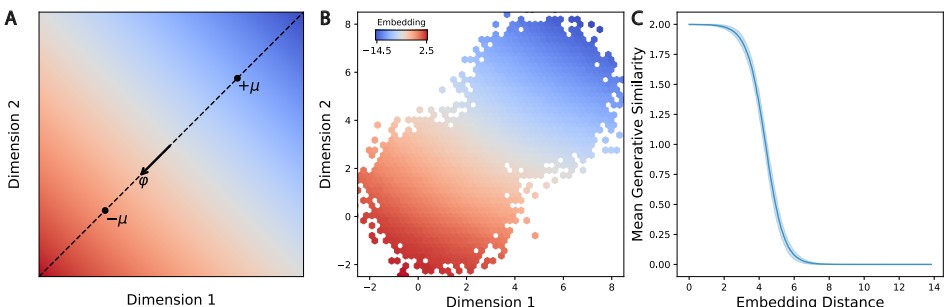

Figure 2: **Encoding the generative similarity of a Gaussian mixture. A.** Optimal linear projection vector $\varphi$ for a symmetric Gaussian mixture with means $\pm\mu$. **B.** Learned 1D embedding values from a two-layer perceptron for points sampled from a 2D Gaussian mixture (colors indicate values). **C.** Mean generative similarity as a function of distance in the embedding space shown in **B** (discretized into 500 quantile bins). Shaded area indicates 95% CIs bootsrapped over data points.

classification test accuracy of 99.7%; additional details in Appendix F). Moreover, we compared distance in the learned embedding space to the theoretical generative similarity values in that case and found excellent agreement (Spearman's $\rho(498) = -0.99$, $p < 10^{-3}$; Figure 2C). We also found that the 95% confidence interval (CI) on the average distance for same pairs was $[1.62, 1.66]$ whereas for different pairs it was $[4.77, 4.91]$, consistent with the prediction of Equation 6.

## 3.2 INSTILLING HUMAN GEOMETRIC SHAPE REGULARITY BIASES

**Background:** Psychological research suggests that the human species is uniquely sensitive to abstract geometric regularity (Henshilwood et al., 2011; Saito et al., 2014). Sablé-Meyer et al. (2021) compared diverse human groups (varying in education, cultural background, and age) to non-human primates on a simple oddball discrimination task. Participants were shown a set of five reference shapes and one "oddball" shape and were prompted to identify the oddball (Figure 3A). The reference shapes were generated using basic geometric regularities: parallel lines, equal sides, equal angles, and right angles, which can be specified by a binary vector corresponding to the presence or absence of these specific geometric features. There were 11 types of quadrilateral reference shapes with varying geometric regularity, from squares (most regular) to random quadrilaterals containing no parallel lines, right angles, or equal angles/sides (least regular; Figure 3B). In each trial, five different versions of the same reference shape (e.g., a square) were shown in different sizes and orientations. The oddball shape was a modified version of the reference shape, in which the lower right vertex was moved such that it violated the regularity of the original reference shape (e.g., moving the lower right vertex of a trapezoid such that it no longer has parallel sides). Figure 3A shows an example trial.

Sablé-Meyer et al. (2021) found that humans were sensitive to these geometric regularities (right angles, parallelism, symmetry, etc.) whereas non-human primates were not. Specifically, they found that human performance was best on the oddball task for the most regular shapes, and systematically decreased as shapes became more irregular. Conversely, non-human primates performed well above chance, but they performed worse than humans overall and, critically, exhibited no influence of geometric regularity (Figure 3B). Additionally, they tested a pretrained convolutional neural network (CNN) model, CorNet (Kubilius et al., 2019), on the task. CorNet (Core Object Recognition Network) is a convolutional neural network model with an architecture that explicitly models the primate ventral visual stream. It is pretrained on a standard supervised object recognition objective on ImageNet and is one of the top-scoring models of "brain-score", a benchmark for testing models of the visual system using both behavioral and neural data (Schrimpf et al., 2018). Like the monkeys, CorNet exhibited no systematic relationship with the level of geometric regularity (Figure 3B).

**Generative Similarity:** Intuitive geometry serves as an ideal case study for our framework because i) it admits a generative similarity measure that can be computed in closed form, and ii) we can use it to test whether our contrastive training framework can induce the human inductive bias observed by Sablé-Meyer et al. (2021) in a neural network. Recall that the shape categories of Sablé-Meyer et al. (2021) can be specified by binary feature vectors corresponding to the presence or absence of abstract

geometric features (equal angles, equal sides, parallel lines, and right angles of the quadrilateral) from which individual examples (or exemplars) can be sampled. Formally, we can define a natural generative process for such shapes as follows: given a set of binary geometric feature variables $\{F_1, \ldots, F_n\} \in \{0, 1\}^n$, we define a hierarchical distribution over shapes by first sampling Bernoulli parameters $\theta_i$ for each feature variable $F_i$ from a prior $\text{Beta}(\alpha, \beta)$, then sampling feature values $f = (f_1, \ldots, f_n)$ from the resulting Bernoulli distributions $\text{Bern}(\theta_i)$, and then uniformly sampling a shape $\sigma(f)$ from a (possibly large) list of available exemplars $\mathcal{S}(f) = \{\sigma_1(f), \ldots, \sigma_M(f)\}$ that are consistent with the sampled feature vector $f$ (the set could also be empty if the geometric features are not realizable due to geometric constraints). In other words, the generative process is defined as

$$\begin{cases} \theta_i \sim \text{Beta}(\alpha, \beta), & \text{sample Bernoulli parameters} \\ f_i \sim \text{Bern}(\theta_i), & \text{sample discrete features} \\ \sigma(f) \sim \text{Uniform}(\mathcal{S}(f)), & \text{sample shape exemplar} \end{cases} \tag{8}$$

This process covers both soft and definite categories, and our current setting corresponds to the special limit $\alpha = \beta \to 0$, in which case the Beta prior over Bernoulli parameters becomes concentrated around 0 and 1 so that the process becomes that of choosing a category specified by a set of geometric attributes and then sampling a corresponding exemplar. In Appendix G, we use the conjugacy relations between the Beta and Bernoulli distributions to derive the generative similarity associated with the process in Equation 8, and we show that in our limit of interest ($\alpha = \beta \to 0$) the corresponding generative similarity measure between shapes $\sigma_1, \sigma_2$ with feature vectors $f_i^{(1)}, f_i^{(2)}$ is given by the formula $\log s(\sigma_1(f^{(1)}), \sigma_2(f^{(2)})) \propto -\sum_i (f_i^{(1)} - f_i^{(2)})^2$.

We used this generative similarity measure to finetune CorNet, the same model Sablé-Meyer et al. (2021) used in their experiments, to see whether our measure would induce the human geometric regularity bias (Figure 3). Specifically, given a random pair of quadrilateral stimuli from Sablé-Meyer et al. (2021), we computed the above quantity (i.e. the Euclidean distance) between their respective binary geometric feature vectors (presence and absence of equal sides, equal angles, and right angles) and finetuned the pretrained CorNet model on a contrastive learning objective using these distances.

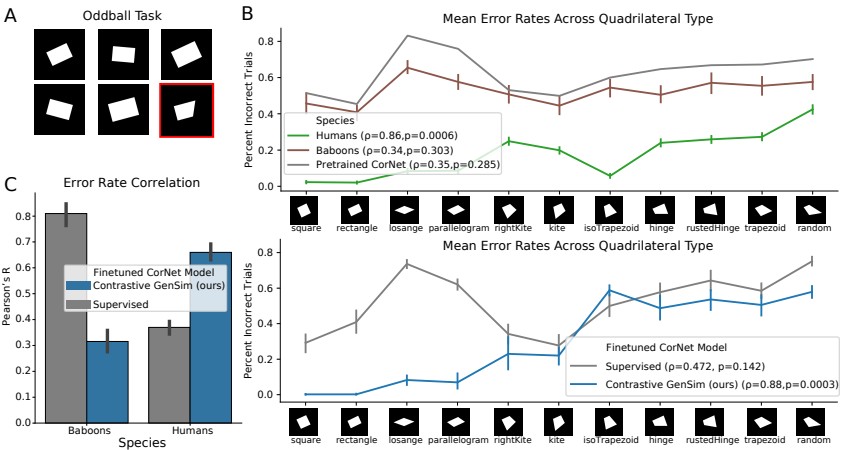

Figure 3: **Generative similarity can instill human geometric regularity biases. A.** The oddball task of Sablé-Meyer et al. (2021) used six quadrilateral stimulus images, in which five images were of the same reference shape (differing in scale and rotation) and one was an oddball (highlighted in red) that diverged from the reference shape's geometric properties. In this example, the reference shape is a rectangle; note that the oddball does not have four right angles like the rectangles. **B.** Sablé-Meyer et al. (2021) examined error rates for humans, monkeys, and pre-trained Convolutional Neural Networks (CNNs) (Kubilius et al., 2019) (top) across quadrilaterals of decreasing geometric regularity. We evaluate the same CNN model with different finetuning objectives (bottom). We report the Spearman rank correlation between model performance and number of geometric regularities across quadrilateral type (see Table 1 in Appendix I for these values). Error bars denote confidence intervals over different subjects (humans, monkeys, or model training seeds). **C.** Correlation between model error rates and human or monkey error rates. Error bars denote 95% CIs over 10 training runs.

This pushed quadrilaterals with similar geometric features together and pulled those with different geometric features apart in the model's representation (additional details regarding training are provided in Appendix I). Like Sablé-Meyer et al. (2021), to test the model on the oddball task, we extract the embeddings for all 6 choice images and choose the oddball as the one that is furthest (i.e. Euclidean distance) from the mean embedding.

**Results:** The geometric regularity effect observed for humans in Sablé-Meyer et al. (2021) was an inverse relationship between geometric regularity and error rate (see green line in top plot of Figure 3B). For example, humans performed best on the most regular shapes, such as squares and rectangles. This regularity effect was absent in the monkey and pretrained CorNet error rates (Figure 3B; top panel). In Figure 3B (bottom panel), we show error rates as a function of geometric regularity on a CorNet model finetuned on generative similarity (GenSim CorNet; blue line bottom plot). We also show the performance of a CorNet model finetuned on a supervised classification objective on the quadrilateral stimuli (grey line bottom plot), where the model must classify which of the 11 categories a quadrilateral belongs to. Note that Sablé-Meyer et al. (2021) also finetuned CorNet on the same supervised classification objective in their supplementary results, and we replicate their results here as a baseline for our proposed method. Both models were trained for 13 epochs (the same number of epochs used by Sablé-Meyer et al. (2021)). Like humans, the model error rates for the GenSim CorNet model significantly increase as the shapes become more irregular (Spearman's $\rho(9) = 0.88, p = 0.0003$). This is not the case for the model finetuned with supervised classification (Spearman's $\rho(9) = 0.472, p = 0.142$). We also correlated the error rates of the finetuned models with those of humans and monkeys (Figure 3C) and see a double dissociation between the two models. Specifically, the generative similarity-trained CorNet model's error rates match human error rates significantly more than monkey error rates, $t(18) = 12.45, p < 0.0001$, whereas those of the baseline supervised CorNet model match monkey error rates significantly more than human error rates (Figure 3C), $t(18) = 17.43, p < 0.0001$. This double dissociation is consistent across different subjects (see Supplementary Figure S3. We replicated the geometric regularity effect when using contrastive learning with generative similarity on a different architecture (Supplementary Figure S1) and also saw that training on a standard contrastive learning objective from SimCLR (Chen et al., 2020) does not yield the regularity effect (Supplementary Figure S2).

### 3.3 LEARNING ABSTRACT DRAWINGS USING GENERATIVE SIMILARITY OVER PROBABILISTIC PROGRAMS

**Background:** Our next test case is based on a recent study on capturing human intuitions of psychological complexity for abstract geometric drawings (Sablé-Meyer et al., 2022). Sablé-Meyer et al. (2022) framed geometric concept learning as probabilistic program induction within the DreamCoder framework (Ellis et al., 2021). A base set of primitives were defined such that motor

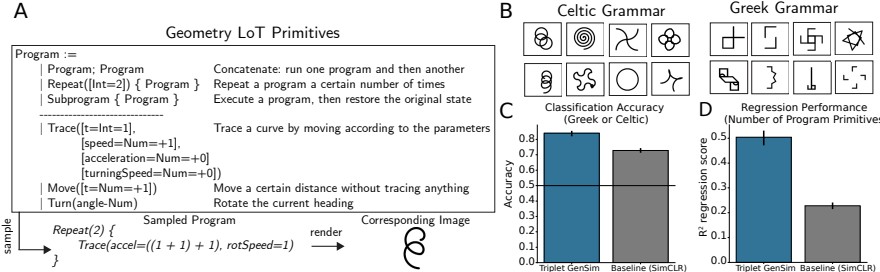

Figure 4: **Generative similarity helps contrastive learning models better represent probabilistic programs. A.** Primitives of the generative Language of Thought (LoT) DreamCoder model implemented in Sablé-Meyer et al. (2022). Primitives are recursively composed to produce symbolic programs that can be rendered into geometric pattern stimuli. **B.** Sablé-Meyer et al. (2022) trained DreamCoder on two sets of drawings, Celtic and Greek, to produce two different grammars that produced qualitatively different drawings. **C.** Performance (with CI's over 10 training runs) of embeddings on classifying images as from the Celtic or Greek grammars. **D.** Performance of embeddings on predicting the number of primitives of the program used to generate the image stimulus (see **A**).

programs that draw geometric patterns are generated through recursive combination of these primitives within a Domain Specific Language (DSL, Figure 4A). The DSL contains motor primitives, such as tracing a particular curve and changing direction, as well as control primitives to recursively combine subprograms, such as $Concat$ (concatenate two subprograms together) and $Repeat$ (repeat a subprogram $n$ times). DreamCoder can be trained on drawings to learn a grammar from which probabilistic programs can be sampled. These probabilistic programs can then be rendered into images such as the ones seen in Figure 4. Sablé-Meyer et al. (2022) used a working memory task with these stimuli to show people's intuitions about the psychological complexity of the image can be modeled through the complexity of the underlying program (Sablé-Meyer et al., 2022).

The study showed that DreamCoder can produce grammars of different abstract drawing styles depending on its training data. For example, when trained on "Greek-style" drawings with highly rectilinear structure, DreamCoder learns a grammar that synthesizes programs which capture this drawing style (Figure 4B). Likewise, when trained on "Celtic-style" drawings that feature lots of circles and curves, DreamCoder learns a different grammar that captures the Celtic drawing style (Figure 4B). Both grammars use the same set of base primitives, but weight the primitives differently and thus produce images that differ in their abstract drawing style.

**Generative Similarity:** To test whether generative similarity over probabilistic programs can allow neural networks to capture the abstract structure that such programs represent we trained a neural network using generative similarity over probabilistic programs from the different grammars discussed in the previous section (Greek or Celtic, see Figure 4B). In this case, the generative similarity is intractable, but we can apply a Monte Carlo approximation to Equation 4 by sampling from the program grammar. We employ the following technique to generate Monte Carlo triplet samples for the triplet contrastive loss function. The anchor is randomly sampled from either the Celtic or Greek grammars that are learned through DreamCoder (with equal probabilities). The positive example is sampled from the same grammar as the anchor and the negative example is another random sample from either the Celtic or Greek grammar with equal probability, consistent with Equation 5.

We used 20k examples from both the Celtic and Greek grammars (40k images in total) for training and 800 examples from each grammar for testing. Because of the similarity of the stimuli in Figure 4B to handwritten characters, we used the same CNN architecture that Snell et al. (2017) used on the Omniglot dataset (Lake et al., 2019) with six convolutional blocks consisting of 64-filter $3 \times 3$ convolution, a batch normalization layer, a ReLU nonlinearity, and a $2 \times 2$ max-pooling layer. As a baseline, we trained another model, with the same architecture and training data, on a standard contrastive learning objective used in the SimCLR paper (Chen et al., 2020). The SimCLR objective produces augmented versions of an image (e.g. random cropping, rotations, gaussian blurring, etc.) with the goal of making representations of an image and its augmented version as similar as possible and representations of different images as dissimilar as possible. See Appendix J for more details.

**Results:** To compare the ability of the contrastive objectives to separate Celtic or Greek images, we took the embeddings of all test images and trained a logistic regression model to classify Celtic or Greek images. Training and evaluation used five-fold cross-validation, tuning the regularization

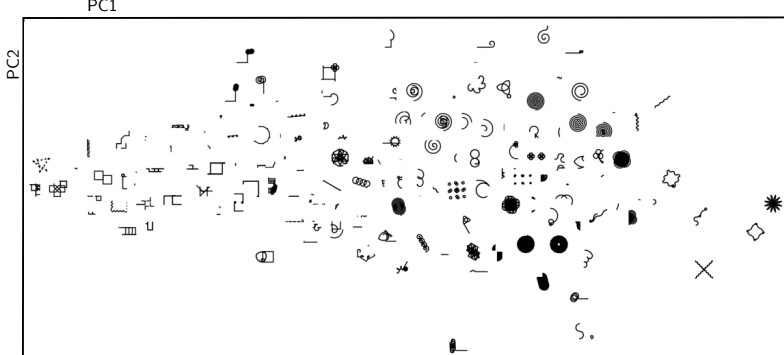

Figure 5: **PCA space of embeddings from model trained with generative similarity triplet loss.** The model clearly seperates "Greek" (left) from "Celtic" (right) styles, with mixed styles in between.

parameter using nested folds within the training set. The mean test set classification accuracy from the embeddings trained with contrastive generative similarity was 84% (95% CI [81.9, 84.9]), significantly higher than that of the SimCLR contrastive learning baseline, 72.8% (95% CI [71.8, 73.8]), $t(18) = 12.81, p < .0001$ (see Figure 4C). We replicated these results on a different architecture (Supplementary Figure S4). We also found that the first two principal components of the model's embeddings can separate the two drawing styles (Figure 5), supporting the notion that generative pretraining facilitates factorization of task-relevant dimensions (Campbell & Cohen, 2024).

We also compared the ability of the learned embeddings to encode properties of the underlying program (Figure 4D). For each test image, we counted the number of motor and control primitives (Figure 4A) within the programs used to generate each image. Then, we trained a ridge regression model to predict this number from the respective image embedding, regressing out average grey-level of the image prior to training as a potential confound. The ridge regression model was trained and evaluated using five-fold cross-validation, where the regularization parameter was tuned using nested folds within the training set. The average test set score using the generative similarity model, $R^2 = 0.50$ (95% CI [0.46, 0.52]), was significantly higher than that of the SimCLR baseline model, $R^2 = 0.23$ (95% CI [0.21, 0.24]), $t(18) = 17.48, p < 0.0001$. This suggests that the embedding space of the generative similarity model better encodes properties of the original programs.

### 3.4 CAPTURING INDUCTIVE BIASES TOWARDS ABSTRACT HIERARCHICAL CATEGORIES IN NATURAL IMAGES WITH GENERATIVE SIMILARITY

**Background:** An important facet of human intelligence is the ability to formulate abstract categories that allow us to generalize across individual stimuli. For example, when encountering a dog for the first time, we may come up with the abstract category of "animal" and generalize this concept to different instances, such as a cat. Humans naturally organize such abstract categories into complex hierarchies (e.g., living organism → animal → mammal → dog; Tenenbaum et al., 2011; Murphy, 2004). This hierarchical structure is reflected in many big machine learning datasets, including ImageNet classes (Deng et al., 2009). This provides a way to test whether generative similarity can capture human-like inductive biases towards abstract categories.

**Generative Similarity:** To show that generative similarity with contrastive learning can enable learning of embeddings that encode abstract categories we use the ImageNet-based dataset "tieredImageNet" (Ren et al., 2018), which organizes ImageNet images into abstract categories that contain multiple subclasses unique to that category (see Figure 6A). Each class belongs to only one category. Additional details are provided in Appendix K.

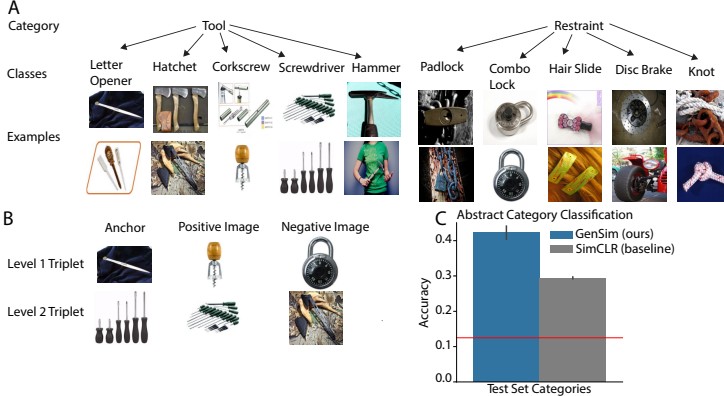

Figure 6: **Generative Similarity for Abstract Categories in Natural Images. A.** The Tiered ImageNet Dataset is organized into high-level, abstract categories that each have their own subclasses. **B.** To train a network on Generative Similarity, we sampled triplets at different levels of abstraction. **C.** Performance of network trained on Generative Similarity vs. SimCLR baseline. The same architecture and training data were used for both. For categories unseen during training, the network trained with Generative Similarity had significantly higher accuracy in predicting image category. Red lines indicate chance levels. Error bars indicate 95% confidence intervals over 5 training runs.

To train on these images with generative similarity, we use a Monte Carlo estimate based on sampling triplets at the two different levels of abstraction in the tieredImageNet dataset (see Figure 6B). At the top level of abstraction, the positive image is sampled from the same category of the anchor image, but with an independent class (so the positive image and anchor image may have different classes). The negative image is from a category sampled independently (so the negative image may have a different category). For the bottom level of abstraction, the positive image is sampled from the same category and same class, and the negative image is sampled from the same category but independent class. To bias the network towards learning higher-level abstract categories, we sampled 80% top-level triplets and 20% bottom-level triplets during training. We used the same triplet-style contrastive loss employed in the abstract drawing style domain (with Euclidean distances). As a baseline, we used SimCLR with image augmentations consisting of random resized crops, random horizontal flips, random gaussian blurring, and random grayscaling of the image.

**Results:** For both losses (Generative Similarity vs. SimCLR), we used a standard ResNet architecture and trained on the training dataset of tieredImageNet for 10 epochs. To quantify how much the learned embeddings have encoded abstract categories, we used the learned model embeddings to train a Linear SVM classifier to classify high-level categories from the images. We did this for images from held-out test categories. Our results, displayed in Figure 6C, show that generative similarity performs significantly better than the baseline at predicting abstract categories, highlighting how generative similarity can capture an abstract notion of high-level tiered categories. We replicated these results on a different architecture (Figure S6).

## 4 DISCUSSION

We have introduced a new framework for learning representations that capture human inductive biases by combining a Bayesian notion of generative similarity with contrastive learning. Our framework is very general and can be applied to any hierarchical generative process, even when the exact form of inferences is intractable, allowing neural networks to capture domains that were previously restricted to symbolic models. To demonstrate the utility and flexibility of our approach we applied it to an array of domains that vary in complexity. First, we investigated an analytically tractable case that involved a mixture of two Gaussians and showed that mean generative similarity monotonically decreases with respect to distance in the contrastively-learned embedding space (Figure 2). Second, we examined a visual perception task with quadrilaterals used in cognitive science (Sablé-Meyer et al., 2021) and showed that our procedure is able to produce the human geometric regularity bias in models that were previously unable to (Figure 3). Third, we show improved representations of probabilistic programs for abstract human drawing styles compared to standard contrastive paradigms like SimCLR (Chen et al., 2020) (Figure 4). Fourth, we use a standard large-scale machine learning dataset, ImageNet, to show generative similarity can learn embeddings that encode abstract, high-level categories compared to standard contrastive paradigms, even those not seen in training (Figure 6).

There are some limitations to our work that point towards future directions. First, there may be multiple candidates reflecting different hypotheses concerning the generative model, and more broadly not all domains admit a transparent hierarchical structure. Characterizing the kind of generative models underlying human judgments in domains of interest (see e.g., Destler et al. (2023) for a recent study on skeletal models of shapes) is key to instilling the right inductive biases in machine models. Therefore, solving the inverse problem, i.e., inferring the generative model itself from the data in tandem with generative similarity is a promising avenue. Second in our work we mainly focus on domains related to vision, but our framework is general enough to be applicable for other modalities. For example, Large Language Models can often produce unpredictable failures in logical (Wan et al., 2024) and causal (Kıcıman et al., 2023) reasoning. Cognitive scientists have written models for human logical reasoning (Piantadosi et al., 2016) or causal learning (Goodman et al., 2011) based on probabilistic Bayesian inference. Potential future work may involve contrastive learning with generative similarity over Bayesian models of reasoning to imbue language models with logical and causal reasoning abilities. Note that, although contrastive learning is most commonly used in vision, there is precedence for its use in the language domain (Gao et al., 2021; Luo et al., 2024).

Strong inductive biases are a hallmark of human intelligence. Finding ways to imbue such biases in machine models is key for developing more generally intelligent AI as well as for achieving human-AI alignment. Our work offers a new path towards that goal.

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

APPENDIX

## A  SEPARATION IN EXPECTATION

Our goal is to show that for any triplet loss function $\ell(\Delta_\phi)$ that is convex and strictly increasing in $\Delta_\phi(x, x^+, x^-) = d_\phi(x, x^+) - d_\phi(x, x^-)$ where $d_\phi(x, y) = d(\phi(x), \phi(y))$ is a given embedding distance measure (e.g., softmax loss or quadratic loss), then the optimal embedding that minimizes Equation 4 ensures that the expected distance between same pairs is strictly smaller than that of different pairs as defined by the generative process in Figure 1A. To see that, let $\phi^*$ denote the optimal embedding and let $\ell^*$ denote its achieved loss. By definition, for any suboptimal embedding $\phi_{\text{sub}}$ which achieves $\ell_{\text{sub}}$ we have $\ell^* < \ell_{\text{sub}}$. One such suboptimal embedding (assuming non-degenerate distributions) is the constant embedding which collapses all samples into a point $\phi_{\text{sub}} = \phi_0$. In that case, we have $\Delta = 0$ and hence $\ell^* < \ell(0)$. Now, using Jensen inequality we have

$$\ell(0) > \ell^* = \mathbb{E}_{p_c}\ell(\Delta_{\phi^*}(X, X^+, X^-)) \geq \ell(\mathbb{E}_{p_c}\Delta_{\phi^*}(X, X^+, X^-)). \tag{A1}$$

Observe next that since $\ell$ is strictly increasing (and hence its inverse is well-defined and strictly increasing) it follows that $\mathbb{E}_{p_c}\Delta_{\phi^*}(X, X^+, X^-) < 0$. Finally, by noting that

$$\mathbb{E}_{p_c}d_{\phi^*}(X, X^+) = \int p(x|\theta^+)p(x^+|\theta^+)p(x^-|\theta^-)p(\theta^+)p(\theta^-)d_{\phi^*}(x, x^+)dxdx^+dx^-d\theta^+d\theta^-$$

$$= \int p(x|\theta^+)p(x^+|\theta^+)p(\theta^+)d_{\phi^*}(x, x^+)dxdx^+d\theta^+$$

$$= \mathbb{E}_{p_{\text{same}}}d_{\phi^*}(X, X^+)$$

and likewise,

$$\mathbb{E}_{p_c}d_{\phi^*}(X, X^-) = \int p(x|\theta^+)p(x^+|\theta^+)p(x^-|\theta^-)p(\theta^+)p(\theta^-)d_{\phi^*}(x, x^-)dxdx^+dx^-d\theta^+d\theta^-$$

$$= \int p(x|\theta^+)p(x^-|\theta^-)p(\theta^+)p(\theta^-)d_{\phi^*}(x, x^-)dxdx^-d\theta^+d\theta^-$$

$$= \mathbb{E}_{p_{\text{diff}}}d_{\phi^*}(X, X^-)$$

we arrive at the desired result

$$\mathbb{E}_{p_{\text{same}}}d_{\phi^*}(X, X^+) < \mathbb{E}_{p_{\text{diff}}}d_{\phi^*}(X, X^-). \tag{A2}$$

## B  GENERATIVE SIMILARITY AS AN OPTIMAL SOLUTION

In what follows we will show that generative similarity (Equation 1) can be derived as the minimizer of the following functional

$$\mathcal{L}[s] = D_{KL}[s||p_{\text{same}}] - D_{KL}[s||p_{\text{diff}}] - \beta^{-1}H(s) + \lambda\left[\int s(x, x')dxdx' - 1\right] \tag{B1}$$

where $D_{KL}$ is the Kullback-Leibler divergence, $H$ is entropy, and the linear integral is a Lagrangian constraint that ensures that $s$ is normalized so that the other terms are well defined. Note that while $\lambda > 0$ is a Lagrange multiplier, $\beta^{-1} > 0$ is a free parameter of our choice that controls the contribution of the entropy term and we may set it to one or a small number if desired. In other words, the minimizer $s^*$ of $\mathcal{L}$ is the maximum-entropy (or entropy-regularized) solution that maximizes the contrast between $p_{\text{same}}$ and $p_{\text{diff}}$ in the $D_{KL}$ sense (i.e. it seeks to assign high weight to pairs with high $p_{\text{same}}$ but low $p_{\text{diff}}$, and low values for pairs with low $p_{\text{same}}$ but high $p_{\text{diff}}$). To derive $s^*$, observe that from the definition of the KL divergence we have

$$D_{KL}[s||p_{\text{same}}] - D_{KL}[s||p_{\text{diff}}] = \int \left[s(x, x')\log\frac{s(x, x')}{p_{\text{same}}(x, x')} - s(x, x')\log\frac{s(x, x')}{p_{\text{diff}}(x, x')}\right]dxdx'$$

$$= \int \left[s(x, x')\log p_{\text{diff}}(x, x') - s(x, x')\log p_{\text{same}}(x, x')\right]dxdx'$$

Next, varying the functional with respect to $s$ we have

$$\frac{\delta \mathcal{L}}{\delta s} = \log p_{\text{diff}}(x_1, x_2) - \log p_{\text{same}}(x_1, x_2) + \beta^{-1} \left[ \log s(x_1, x_2) + 1 \right] + \lambda = 0 \quad \text{(B2)}$$

which yields

$$s^*(x_1, x_2) = \frac{1}{Z(\beta, \lambda)} \left[ \frac{p_{\text{same}}(x_1, x_2)}{p_{\text{diff}}(x_1, x_2)} \right]^\beta \quad \text{(B3)}$$

where we defined $Z(\beta, \lambda) \equiv e^{\beta \lambda + 1}$. Next, from the Lagrange multiplier equation $\delta_\lambda \mathcal{L} = 0$ we have

$$Z(\beta, \lambda) = \int \left[ \frac{p_{\text{same}}(x_1, x_2)}{p_{\text{diff}}(x_1, x_2)} \right]^\beta dx_1 dx_2 \quad \text{(B4)}$$

which fixes $\lambda$ as a function of $\beta$ assuming that the right-hand integral converges. Two possible sources of divergences are i) $p_{\text{diff}}(x_1, x_2)$ approaches zero while $p_{\text{same}}(x_1, x_2)$ remains finite, and ii) the integral is carried over an unbounded region without the ratio decaying fast enough. The latter issue can be resolved by simply assuming that the space is large but bounded and that the main probability mass of the generative model is far from the boundaries (which is plausible for practical applications). As for the former, observe that when $p_{\text{diff}}(x_1, x_2) \equiv \int p(x_1|\theta_1) p(x_2|\theta_2) p(\theta_1) p(\theta_2) d\theta_1 d\theta_2 = 0$ it implies (from non-negativity) that $p(x_1|\theta_1) p(x_2|\theta_2) = 0$ for all $\theta_{1,2}$ in the support of $p(\theta)$ which in turn implies that $p_{\text{same}}(x_1, x_2) \equiv \int p(x_1|\theta) p(x_2|\theta) p(\theta) d\theta = 0$. In other words, if $p_{\text{different}}$ vanishes then so does $p_{\text{same}}$ (but not vice versa, e.g. if $p(x_1|\theta)$ and $p(x_2|\theta)$ have non-overlapping support as a function of $\theta$). Likewise, the rate at which these approach zero is also controlled by the same factor $p(x_1|\theta_1) p(x_2|\theta_2) \to 0$ and so we expect the ratio to be generically well-behaved.

Finally, setting $\beta = 1$ we arrive at the desired Bayes odds relation

$$s^*(x_1, x_2) \propto \frac{p_{\text{same}}(x_1, x_2)}{p_{\text{diff}}(x_1, x_2)} = \frac{p(\text{same}|x_1, x_2)}{p(\text{different}|x_1, x_2)} \quad \text{(B5)}$$

where the second equality follows from the fact that we assumed that *a priori* $p(\text{same}) = p(\text{different})$. As a sanity check of the convergence assumptions, consider the case of a mixture of two one-dimensional Gaussians with means $\mu_1 = -\mu_2 = \mu$ and uniform prior, and as a test let us set $\sigma = 1$ and $\mu \gg 1$ so that the Gaussians do not overlap and are far from the origin. In this case, we assume that the space is finite $x \in [-\Lambda, \Lambda]$ such that $\Lambda \gg \mu \gg 1$ so that the Gaussians are unaffected by the boundary. Then, for points that are far from the Gaussian centers, e.g. at the origin $x_1 = x_2 = 0$ for which the likelihoods are exponentially small we have

$$\frac{p(x_1, x_2 | \text{same})}{p(x_1, x_2 | \text{different})} = \frac{\frac{1}{2} e^{-\frac{(+\mu)^2}{2}} \times e^{-\frac{(+\mu)^2}{2}} + \frac{1}{2} e^{-\frac{(-\mu)^2}{2}} \times e^{-\frac{(-\mu)^2}{2}}}{(\frac{1}{2} e^{-\frac{(+\mu)^2}{2}} + \frac{1}{2} e^{-\frac{(-\mu)^2}{2}}) \times (\frac{1}{2} e^{-\frac{(+\mu)^2}{2}} + \frac{1}{2} e^{-\frac{(-\mu)^2}{2}})} = 1 \quad \text{(B6)}$$

which is indeed finite.

## C    CONNECTIONS TO OTHER LOSS FUNCTIONS

The unregularized divergence difference $D_{KL}[s||p_{\text{same}}] - D_{KL}[s||p_{\text{diff}}]$ can also be related to a special case of the loss objective in Equation 4. Specifically, observe that

$$D_{KL}[s||p_{\text{same}}] - D_{KL}[s||p_{\text{diff}}] = \int [s(x, x') \log p_{\text{diff}}(x, x') - s(x, x') \log p_{\text{same}}(x, x')] dx dx' \quad \text{(C1)}$$

where $D_{KL}[p||q] = \int p(x) \log[p(x)/q(x)] dx$ is the Kullback-Leibler (KL) divergence. While the left-hand-side in Equation C1 may seem rather different from Equation 4, the cancellation in the KL divergences yields a special case of Equation 4 with $\ell(\Delta) = \Delta$ upon minimal redefinitions, namely, recasting distance measures as similarities $d(x, y) \to s_0 - s(x, y)$ and substituting probabilities with their logarithm $p \to \log p$ (i.e., applying a monotonic transformation; see Appendix D). Indeed, varying the functional in Equation 4 with respect to $s$ along with a simple quadratic regularizer (see Appendix D) yields $s^*(x_1, x_2) \propto p_{\text{same}}(x_1, x_2) - p_{\text{diff}}(x_1, x_2)$ which is equivalent to the generative similarity measure (Equation 1) up to a monotonic transformation of probabilities $p \to \log p$.

# D   THE SPECIAL CASE OF $\ell(\Delta) = \Delta$

We consider the triplet loss objective under the special case of $\ell(\Delta) = \Delta$ where $\Delta(x, x^+, x^-) = d(x, x^+) - d(x, x^-)$. Recasting the distance measures as similarities $d(x, y) \to s_0 - s(x, y)$ and unpacking Equation 4 we have

$$
\begin{aligned}
\mathcal{L}[s] &= \mathbb{E}_{p_c} \Delta(X, X^+, X^-) \\
&= \mathbb{E}_{p_c} s(X, X^-) - \mathbb{E}_{p_c} s(X, X^+) \\
&= \mathbb{E}_{p_{\mathrm{diff}}} s(X, X^-) - \mathbb{E}_{p_{\mathrm{same}}} s(X, X^+) \\
&= \int [s(x, x') p_{\mathrm{diff}}(x, x') - s(x, x') p_{\mathrm{same}}(x, x')] dx dx'
\end{aligned}
$$

where the third equality follows from an identical derivation to the one found in Appendix A above Equation A2.

Our goal next is to find the similarity function which minimizes $\mathcal{L}[s]$ by varying it with respect to $s$, i.e., $\delta_s \mathcal{L} = 0$. As before, since $\mathcal{L}$ is linear in $s$ we need to add a suitable regularizer to derive a solution (otherwise $\delta_s \mathcal{L} = 0$ has no solutions). Here we are no longer committed to a probabilistic interpretation of $s$ and so a natural choice would be a quadratic regularizer

$$
\mathcal{L}_{\mathrm{reg}}[s] = \mathcal{L}[s] + \lambda \left( \int s^2(x, x') dx dx' - \Lambda \right) \tag{D1}
$$

for some constants $\Lambda, \lambda > 0$. Varying the Lagrangian with respect to the similarity measure we have

$$
\frac{\delta \mathcal{L}_{\mathrm{reg}}}{\delta s} = p_{\mathrm{diff}}(x_1, x_2) - p_{\mathrm{same}}(x_1, x_2) + 2\lambda s(x_1, x_2) = 0 \tag{D2}
$$

This in turn implies that the optimal similarity measure is given by

$$
s^*(x_1, x_2) = \frac{1}{2\lambda} \left[ p_{\mathrm{same}}(x_1, x_2) - p_{\mathrm{diff}}(x_1, x_2) \right] \tag{D3}
$$

Likewise, for the Lagrange multiplier we have

$$
\frac{\delta \mathcal{L}_{\mathrm{reg}}}{\delta \lambda} = \int s^2(x, x') dx dx' - \Lambda = 0 \tag{D4}
$$

Plugging in the optimal solution we have

$$
\frac{1}{4\lambda^2} \int \left[ p_{\mathrm{same}}(x, x') - p_{\mathrm{diff}}(x, x') \right]^2 dx dx' - \Lambda = 0 \tag{D5}
$$

The integral is positive since it is the squared difference between two normalized probability distributions, and so denoting its value as $C_p > 0$ we can solve for $\lambda$

$$
\lambda = \frac{1}{2} \sqrt{\frac{C_p}{\Lambda}} \tag{D6}
$$

Thus, putting everything together we have

$$
s^*(x_1, x_2) = \sqrt{\frac{\Lambda}{C_p}} \left[ p_{\mathrm{same}}(x_1, x_2) - p_{\mathrm{diff}}(x_1, x_2)) \right] \tag{D7}
$$

# E   GAUSSIAN MIXTURES AND LINEAR PROJECTIONS

To derive the linear projection result, we start by plugging in the definition of the Gaussian mixture generative process (i.e., uniformly sampling a Gaussian and then sampling points from it) into Equation 4

$$
\mathbb{E}_{p_c} \ell(X, X^+, X^-) = \sum_{i=1,2} \sum_{j=1,2} \int \ell(\phi(x), \phi(x^+), \phi(x^-)) \times
$$

$$
\times \frac{1}{4} \frac{1}{((2\pi)^d \sigma^{2d})^{3/2}} \exp \left( -\frac{(x - \mu_i)^2 + (x^+ - \mu_i)^2 + (x^- - \mu_j)^2}{2\sigma^2} \right) dx dx^+ dx^-
$$

Now, recall that

$$\begin{aligned}
\ell(x, x^+, x^-) &= \phi_\varphi(x) \cdot \phi_\varphi(x^-) - \phi_\varphi(x) \cdot \phi_\varphi(x^+) \\
&= (\varphi \cdot x)(\varphi \cdot x^-) - (\varphi \cdot x)(\varphi \cdot x^+)
\end{aligned}$$

Substituting into the loss formula and integrating we have

$$\begin{aligned}
\mathbb{E}_{p_c}\ell \propto &\sum_{i,j} \int (\varphi \cdot x)(\varphi \cdot x^-) \exp\left(-\frac{(x-\mu_i)^2 + (x^- - \mu_j)^2}{2\sigma^2}\right) dx dx^- \\
&- 2\sum_i \int (\varphi \cdot x)(\varphi \cdot x^+) \exp\left(-\frac{(x-\mu_i)^2 + (x^+ - \mu_i)^2}{2\sigma^2}\right) dx dx^+
\end{aligned}$$

Note next that since $x^+$ and $x^-$ are dummy integration variables, we can further rewrite

$$\begin{aligned}
\mathbb{E}_{p_c}\ell \propto &\sum_{i \neq j} \int (\varphi \cdot x)(\varphi \cdot x^-) \exp\left(-\frac{(x-\mu_i)^2 + (x^- - \mu_j)^2}{2\sigma^2}\right) dx dx^- \\
&- \sum_i \int (\varphi \cdot x)(\varphi \cdot x^+) \exp\left(-\frac{(x-\mu_i)^2 + (x^+ - \mu_i)^2}{2\sigma^2}\right) dx dx^+
\end{aligned}$$

Thus, using the fact that the distributions are separable and standard Gaussian moment formulae we arrive at

$$\mathbb{E}_{p_c}\ell \propto \sum_{i \neq j}(\varphi \cdot \mu_i)(\varphi \cdot \mu_j) - \sum_i (\varphi \cdot \mu_i)(\varphi \cdot \mu_i) \tag{E1}$$

Finally, plugging in $\mu_1 = -\mu_2 = \mu$ and using the fact that $||\varphi||_2^2 = \varphi \cdot \varphi = 1$ we have

$$\mathbb{E}_{p_c} \propto -4(\varphi \cdot \mu)^2 = -4||\mu||_2^2 \cos^2 \theta_{\varphi\mu}. \tag{E2}$$

## F  GAUSSIAN MIXTURES AND TWO-LAYER PERCEPTRONS

To test the Monte Carlo approximation of Equation 5 and to see how well it tracks the theoretical generative similarity, we considered an embedding family that is parametrized by two-layer perceptrons. For the generative family, we chose as before a mixture of two Gaussians, this time with mean values of $\mu_1 = (5, 5)$ and $\mu_2 = (1, 1)$ and unit variance $\sigma^2 = 1$. As for the loss function, here we used a quadratic (Euclidean) loss of the form

$$\mathcal{L} = \frac{1}{N_{\text{triplets}}} \sum_{\{x, x^+, x^-\}} \left[ (\phi(x) - \phi(x^+))^2 - (\phi(x) - \phi(x^-))^2 \right] \tag{F1}$$

where $\{x, x^+, x^-\}$ are triplets sampled from Equation 5. We trained the perceptron model using 10,000 triplets (learning rate $= 10^{-5}$, hidden layer size $= 32$, batch-size $= 256$, and 300 epochs). The resulting model successfully learned to distinguish the two Gaussians as seen visually from the embedding values in Figure 2B, and also from the test accuracy of 99.7%. Finally, we wanted to see how well the embedding distance tracked the theoretical generative similarity, which in this case can be derived in closed form by simply plugging in the Gaussian distributions in Equation 1

$$s(x_1, x_2) = \frac{\frac{1}{2}e^{-\frac{(x_1-\mu_1)^2}{2\sigma^2}} \times e^{-\frac{(x_2-\mu_1)^2}{2\sigma^2}} + \frac{1}{2}e^{-\frac{(x_1-\mu_2)^2}{2\sigma^2}} \times e^{-\frac{(x_2-\mu_2)^2}{2\sigma^2}}}{\left(\frac{1}{2}e^{-\frac{(x_1-\mu_1)^2}{2\sigma^2}} + \frac{1}{2}e^{-\frac{(x_1-\mu_2)^2}{2\sigma^2}}\right) \times \left(\frac{1}{2}e^{-\frac{(x_2-\mu_1)^2}{2\sigma^2}} + \frac{1}{2}e^{-\frac{(x_2-\mu_2)^2}{2\sigma^2}}\right)}. \tag{F2}$$

The mean generative similarity as a function of embedding distance between pairs (grouped into 500 quantile bins) is shown in Figure 2C. We see that it is indeed a monotonically decreasing function of distance in embedding space (Spearman's $\rho(498) = -0.99$, $p < 10^{-3}$). Moreover, we found that the average distance 95% (1.96-sigma) confidence interval (CI) for same pairs was $[1.62, 1.66]$ whereas for different pairs it was $[4.77, 4.91]$, consistent with the prediction of Equation 6.

# G   GENERATIVE SIMILARITY OF GEOMETRIC SHAPE DISTRIBUTIONS

Our goal is to derive the generative similarity measure associated with the process in Equation 8

$$s(\sigma_1(f^{(1)}), \sigma_2(f^{(2)})) = \frac{p_{\text{same}}(\sigma_1, \sigma_2)}{p_{\text{diff}}(\sigma_1, \sigma_2)} \tag{G1}$$

Plugging in the different Beta, Bernoulli, and uniform distributions into the nominator of Equation 1 we have

$$p_{\text{same}}(\sigma_1, \sigma_2) = \sum_{\hat{f}_{1\cdots n}^{(1,2)}} \int \frac{\delta(\hat{f}^{(1)} - f^{(1)})}{|\mathcal{S}(\hat{f}^{(1)})|} \frac{\delta(\hat{f}^{(2)} - f^{(2)})}{|\mathcal{S}(\hat{f}^{(2)})|}$$

$$\times \prod_i \left[ \theta_i^{f_i^{(1)} + f_i^{(2)} + \alpha - 1} (1 - \theta_i)^{\bar{f}_i^{(1)} + \bar{f}_i^{(2)} + \beta - 1} / \mathrm{B}(\alpha, \beta) \right] d\theta_{1\cdots n}$$

where we defined $\bar{f}_i = 1 - f_i$ and used the definition of the Bernoulli distribution $\mathrm{Bern}(f_i; \theta_i) = \theta_i^{f_i}(1 - \theta_i)^{\bar{f}_i}$, and the Beta distribution $\mathrm{Beta}(\theta_i; \alpha, \beta) = \theta_i^{\alpha - 1}(1 - \theta_i)^{\beta - 1} / \mathrm{B}(\alpha, \beta)$ where B is the Beta function and is given by $\mathrm{B}(z_1, z_2) = \int_0^1 dt\, t^{z_1 - 1}(1 - t)^{z_2 - 1}$ which is well-defined for all positive numbers $z_1, z_2 > 0$. Note that the delta function $\delta(\hat{f} - f)$ simply enforces the fact that by definition each stimulus is consistent with only one set of feature values (otherwise there would be at least one feature of the stimulus that is both True and False which is a contradiction). Likewise, $|\mathcal{S}(\hat{f})|$ is the cardinality of the exemplar set associated with the feature vector $\hat{f}$ which accounts for uniform sampling. Likewise, for the denominator of Equation 1 we have

$$p_{\text{diff}}(\sigma_1, \sigma_2) = \sum_{\hat{f}_{1\cdots n}^{(1)}} \int \frac{\delta(\hat{f}^{(1)} - f^{(1)})}{|\mathcal{S}(\hat{f}^{(1)})|} \prod_i \left[ \theta_{(1)i}^{f_i^{(1)} + \alpha - 1} (1 - \theta_{(1)i})^{\bar{f}_i^{(1)} + \beta - 1} / \mathrm{B}(\alpha, \beta) \right] d\theta_{1\cdots n}^{(1)}$$

$$\times \sum_{\hat{f}_{1\cdots n}^{(2)}} \int \frac{\delta(\hat{f}^{(2)} - f^{(2)})}{|\mathcal{S}(\hat{f}^{(2)})|} \prod_j \left[ \theta_{(2)j}^{f_j^{(2)} + \alpha - 1} (1 - \theta_{(2)j})^{\bar{f}_j^{(2)} + \beta - 1} / \mathrm{B}(\alpha, \beta) \right] d\theta_{1\cdots n}^{(2)}$$

The above integrals might seem quite complicated at first but the conjugacy relation between the Beta and Bernoulli distributions as well as the delta functions simplify things drastically. Indeed, the delta functions cancel the summation over features, and the cardinality factors cancel out in the ratio so that we are left with a collection of Beta function factors (see definition of Beta function above)

$$s(\sigma_1(f^{(1)}), \sigma_2(f^{(2)})) = \frac{\prod_i \mathrm{B}(f_i^{(1)} + f_i^{(2)} + \alpha, \bar{f}_i^{(1)} + \bar{f}_i^{(2)} + \beta)\mathrm{B}(\alpha, \beta)}{\prod_i \mathrm{B}(f_i^{(1)} + \alpha, \bar{f}_i^{(1)} + \beta) \prod_j \mathrm{B}(f_j^{(2)} + \alpha, \bar{f}_j^{(2)} + \beta)} \tag{G2}$$

Taking the logarithm and rearranging the terms we have

$$\log s(\sigma_1(f^{(1)}), \sigma_2(f^{(2)})) = \sum_i \log \frac{\mathrm{B}(f_i^{(1)} + f_i^{(2)} + \alpha, \bar{f}_i^{(1)} + \bar{f}_i^{(2)} + \beta)\mathrm{B}(\alpha, \beta)}{\mathrm{B}(f_i^{(1)} + \alpha, \bar{f}_i^{(1)} + \beta)\mathrm{B}(f_i^{(2)} + \alpha, \bar{f}_i^{(2)} + \beta)} \tag{G3}$$

Now, recall the following Beta function identities[1]

$$\mathrm{B}(x + 1, y) = \frac{x}{x + y}\mathrm{B}(x, y); \quad \mathrm{B}(x, y + 1) = \frac{y}{x + y}\mathrm{B}(x, y) \tag{G4}$$

Using these identities we can group and simplify the different ratios contributing to the sum depending on the values of the features. If $f_i^{(1)} = f_i^{(2)} = 1$ then we have

$$\log \frac{\mathrm{B}(2 + \alpha, 0 + \beta)\mathrm{B}(\alpha, \beta)}{\mathrm{B}(1 + \alpha, 0 + \beta)\mathrm{B}(1 + \alpha, 0 + \beta)} = \log \frac{\alpha + 1}{\alpha + \beta + 1} \frac{\alpha + \beta}{\alpha} \tag{G5}$$

---

[1]These follow from the fact that $\mathrm{B}(z_1, z_2) = \Gamma(z_1)\Gamma(z_2)/\Gamma(z_1 + z_2)$ where $\Gamma$ is the Gamma function which satisfies $\Gamma(z + 1) = z\Gamma(z)$ for any $z > 0$ (Artin, 2015).

If on the other hand $f_i^{(1)} = 1$ and $f_i^{(2)} = 0$ or $f_i^{(1)} = 0$ and $f_i^{(2)} = 1$ then we have

$$\log \frac{\mathrm{B}(1+\alpha, 1+\beta)\mathrm{B}(\alpha,\beta)}{\mathrm{B}(1+\alpha, 0+\beta)\mathrm{B}(0+\alpha, 1+\beta)} = \log \frac{\beta}{\alpha+\beta+1}\frac{\alpha+\beta}{\beta} \tag{G6}$$

and finally for $f_i^{(1)} = f_i^{(2)} = 0$ we have

$$\log \frac{\mathrm{B}(0+\alpha, 2+\beta)\mathrm{B}(\alpha,\beta)}{\mathrm{B}(0+\alpha, 1+\beta)\mathrm{B}(0+\alpha, 1+\beta)} = \log \frac{\beta+1}{\alpha+\beta+1}\frac{\alpha+\beta}{\beta} \tag{G7}$$

Next, defining $\Sigma_1$ and $\Sigma_2$ to be the sets of features that hold true for stimuli $\sigma_1$ and $\sigma_2$, we can write

$$\log s(\sigma_1(f^{(1)}), \sigma_2(f^{(2)})) = |\Sigma_1 \cap \Sigma_2| \log \frac{\alpha+1}{\alpha+\beta+1}\frac{\alpha+\beta}{\alpha}$$
$$+ (|\Sigma_1 - \Sigma_2| + |\Sigma_2 - \Sigma_1|) \log \frac{\beta}{\alpha+\beta+1}\frac{\alpha+\beta}{\beta}$$
$$+ |\bar{\Sigma}_1 \cap \bar{\Sigma}_2| \log \frac{\beta+1}{\alpha+\beta+1}\frac{\alpha+\beta}{\beta}$$

where $|\Sigma_1 \cap \Sigma_2|$ is the number features that hold true for both stimuli, $|\Sigma_i - \Sigma_j|$ is the number of features that hold true for $\sigma_i$ but not for $\sigma_j$, and finally $|\bar{\Sigma}_1 \cap \bar{\Sigma}_2|$ is the number of features that hold neither for $\sigma_1$ nor for $\sigma_2$. Observe next that by definition $|\bar{\Sigma}_1 \cap \bar{\Sigma}_2| = n - |\Sigma_1 \cap \Sigma_2| - |\Sigma_1 - \Sigma_2| - |\Sigma_2 - \Sigma_1|$ where $n$ is the overall number of features. From here it follows that

$$\log s(\sigma_1(f^{(1)}), \sigma_2(f^{(2)})) = |\Sigma_1 \cap \Sigma_2| \log \frac{\alpha+1}{\alpha}\frac{\beta}{\beta+1}$$
$$+ (|\Sigma_1 - \Sigma_2| + |\Sigma_2 - \Sigma_1|) \log \frac{\beta}{\beta+1}$$
$$+ n \log \frac{\beta+1}{\alpha+\beta+1}\frac{\alpha+\beta}{\beta}$$

In the limit of $\alpha = \beta \to 0$ we have

$$\log s(\sigma_1(f^{(1)}), \sigma_2(f^{(2)}) = n \log 2 - \log \frac{\beta+1}{\beta}(|\Sigma_1 - \Sigma_2| + |\Sigma_2 - \Sigma_1|) \tag{G8}$$

where the first term is simply a constant. Finally, observe that

$$|\Sigma_1 - \Sigma_2| + |\Sigma_2 - \Sigma_1| = \sum_i f_i^{(1)}(1 - f_i^{(2)}) + f_i^{(2)}(1 - f_i^{(1)}) \tag{G9}$$
$$= \sum_i f_i^{(1)} - 2f_i^{(1)}f_i^{(2)} + f_i^{(2)}$$
$$= \sum_i (f_i^{(1)} - f_i^{(2)})^2$$

where the third equality follows from the fact that $f^2 = f$ for binary features. In other words, the generative similarity reduces to a monotonically decreasing function of the Euclidean distance between the geometric features of shapes

$$\log s(\sigma_1(f^{(1)}), \sigma_2(f^{(2)})) = n \log 2 - \log \frac{\beta+1}{\beta} \sum_i (f_i^{(1)} - f_i^{(2)})^2 \tag{G10}$$

which is the desired result.

## H    REPRODUCIBILITY

The authors of this work have made great efforts in ensuring its reproducibility. Anonymized code to train the models used in Figures 2–6 are included in the attached file to the submission. Detailed proofs of theoretical results mentioned in this work are in Appendices A–G. Details of our empirical experiments are in Appendices I–K. We include multiple results for control experiments in Appendices L–Q, including reproducing major results with different architectures.

Table 1: List of geometric regularities for each quadrilateral type (sorted from most regular to least)

| shape | rightAngles | parallels | symmetry | equalSides | equalAngles |
|---|---|---|---|---|---|
| square | 4 | 2 | 4 | 4 | 4 |
| rectangle | 4 | 2 | 2 | 2 | 4 |
| losange | 0 | 0 | 2 | 4 | 2 |
| parallelogram | 0 | 2 | 1 | 2 | 2 |
| rightKite | 2 | 0 | 1 | 2 | 2 |
| kite | 0 | 0 | 1 | 2 | 2 |
| isoTrapezoid | 0 | 1 | 1 | 1 | 2 |
| hinge | 1 | 0 | 0 | 1 | 0 |
| rustedHinge | 0 | 0 | 0 | 1 | 0 |
| trapezoid | 0 | 1 | 0 | 0 | 0 |
| random | 0 | 0 | 0 | 0 | 0 |

## I    DETAILS ON QUADRILATERAL EXPERIMENT

For our main experiments, we use the CorNet model which was used in the original work (variant 'S') that introduces the Oddball task (Sablé-Meyer et al., 2021). CorNet contains four "areas" corresponding to the areas of the visual stream: V1, V2, V4, and IT. Each area contains convolutional and max pooling layers. There are also biologically plausible recurrent connections between areas (e.g., V4 to V1). After IT, the penultimate area in the visual stream, a linear layer is used to readout object categories. The model is pretrained on ImageNet on a standard supervised object recognition objective. The pretrained CorNet model's performance on the Oddball task is reported in Figure 3B.

For finetuning the model on the supervised classification objective, we followed the protocol used in the supplementary results of Sablé-Meyer et al. (2021). Specifically, 11 new object categories are added to the model's last layer, and the model is trained to classify a quadrilateral as one of the 11 categories shown in Figure 3B. For training data, we used quadrilaterals from all 11 categories with different scales and rotations (though the specific quadrilateral images used in the test trials were held out). We used a learning rate of 5e-6 using the Adam optimizer with a cross entropy loss. Training was conducted on an NVIDIA Quadro P6000 GPU with 25GB of memory.

For finetuning the model on the generative similarity contrastive objective, we first calculated the Euclidean distance of the model's final layer embedding between different quadrilateral images, then calculated the Euclidean distance between the quadrilaterals' respective geometry feature vectors, and finally used the mean squared error between the embeddings' distance and the feature vectors' distance as the loss. The geometric feature vectors were a set of 22 binary features encoding the following properties: 6 features per pair of edges encoding whether their lengths are equal or not, 6 features per pair of angles coding whether their angles are equal or not, 6 features per pair of edges encoding whether they were parallel or not, and 4 features per angle encoding whether they were right angles or not. See Table 1 for a list of these values. Like the supervised model, we used training data from each category of quadrilaterals with different scales and rotations (though specific images used in the test trials were held out). We used the Adam optimizer with a learning rate of 5e-4. We used the exact same training data, learning rate, and optimizer when running the control experiments for finetuning CorNet on the SimCLR objective (Figure S2). Training was conducted on an NVIDIA Quadro P6000 GPU with 25GB of memory.

## J    DETAILS ON DRAWING STYLES EXPERIMENT

We used the DreamCoder grammars Sablé-Meyer et al. (2022) trained on Greek and Celtic drawings respectively to obtain training data. Both models used the same DSL (see Figure 4A) but, because they are trained on different images, they weigh those primitives differently and thus combine primitives differently when sampling from the grammar. We obtained 20k images from both grammars (40k images in total) and used 800 additional examples from each grammar for testing. Each image as a $128 \times 128$ gray-scale image. Images were normalized to have pixel values between 0-1 by dividing by 255.

Because of the similarity of the stimuli in Figure 4B to handwritten characters, we used the same CNN architecture that Snell et al. (2017) used on the Omniglot dataset (Lake et al., 2019) with six convolutional blocks consisting of 64-filter $3 \times 3$ convolution, a batch normalization layer, a ReLU nonlinearity, and a $2 \times 2$ max-pooling layer. This network outputs a 256-dimensional embedding. Our experiment was replicated with another CNN architecture (CorNet), which yielded similar results (Figure S4). We used two different training objectives: a standard contrastive learning objective from SimCLR (Chen et al., 2020) and one based on a Monte-Carlo estimate of generative similarity. For both objectives, we used the same learning rate 1e-3 and the same Adam optimizer with a batch size of 128.

For the SimCLR baseline objective, images in the batch were randomly augmented. The augmentations were: random resize crop, random horizontal flips, and random Gaussian blurs. The original SimCLR paper also had augmentations corresponding to color distortions which we did not use because our data were already grayscale images. Like SimCLR, we used the InfoNCE loss function (Oord et al., 2018). Let $v_i$ be the embedding of image $i$ and $v_i'$ be the embedding of image $i$'s augmented counterpart. The loss is $\frac{1}{N} \sum_{i=1}^{N} \log \frac{f(v_i, v_i')}{\frac{1}{N} \sum_j \exp f(v_i, v_j')}$ where $f$ is a similarity function between embeddings (SimCLR used cosine similarity). This effectively pushes representations of images and their augmented counterparts to be more similar while also pushing representations of images and other images' augmented counterparts to be more dissimilar. Training was conducted with one NVIDIA Tesla P100 GPU with 16GB of memory.

For the generative similarity objective, let image $a$ be the representation of the anchor image that is sampled from a random grammar $c \in \{\text{Celtic}, \text{Greek}\}$. Let $p$ be the representation of a positive image that is another image sampled from $c$ and $n$ be the negative image that is randomly sampled from either grammar (and is therefore considered an independent sample). Using these samples, we compute the positive Euclidean distance $d(p, a)$ and the negative Euclidean distance $d(n, a)$ in order to compute the loss function $d(p, a) - d(n, a)$ (see Equation 5). Training was conducted with one NVIDIA Tesla P100 GPU with 16GB of memory.

## K  DETAILS ON ABSTRACT CATEGORIES EXPERIMENT

Our training data came from the tieredImageNet dataset (Ren et al., 2018). In the training set, there are 20 categories containing a total of 351 classes (in which each class can contain thousands of examples). In the test set, there are 8 categories containing a total of 151 classes. The categories are mutually exclusive so no class belongs to more than one category. Images were resized to $224 \times 224 \times 3$ (for 3 RGB channels) before training. We trained a standard ResNet (He et al., 2016) architecture on both types of losses. We used two different training objectives: a standard contrastive learning objective from SimCLR (Chen et al., 2020) and one based on a Monte-Carlo estimate of generative similarity. For both objectives, we used the same learning rate 1e-3 and the same Adam optimizer with a batch size of 256.

For the SimCLR objective, we used the same objective and image augmentations as the drawing style experiment, but also included random grayscaling since ImageNet images are colored. Training was conducted with one NVIDIA Tesla P100 GPU with 16GB of memory.

For the generative similarity objective, we used the same Euclidean-based distance for the triplet loss as the Drawing Styles experiment. However, we included a new mechanism for identifying "high-level" and "low-level" triplets (see Figure 6B). For high-level triplets, the idea is that we are sampling from the top-level in the hierarchical generative model (Figure 6A). The positive image is sampled from the same category of the anchor image, but with an independent class (so the positive image and anchor image may have different classes). The negative image is from a category sampled independently (so the negative image may have a different category). This corresponds to a sample from the generative model at the top of the hierarchy without consideration for the bottom level (the classes). For low-level triplets, we are sampling from the bottom-level of the hierarchical generative model, conditioned on a specific high-level category. Therefore, the positive image is sampled from the same category and same class, and the negative image is sampled from the same category but independent class. To bias the network towards learning higher-level abstract categories (as humans typically do), we sampled $80\%$ top-level triplets and $20\%$ bottom-level triplets during training. Training was conducted with one NVIDIA Tesla P100 GPU with 16GB of memory.

For evaluating the ability of the learned embeddings to encode information about abstract categories, we fit a Linear SVM to predict either images from previously unseen test categories (8-way classification). This analysis was also repeated for predicting the training categories (20-way classification) as a supplementary analysis (Figure S5). The SVM was fit for upto 100 epochs (stopping when the loss fails to decrease by more than 1e-3 for 5 epochs). We trained the SVM within a three-fold cross validation loop, evaluating classification accuracy on a held-out test set, and then reported the mean across the three folds.

## L  REPRODUCTION OF GEOMETRIC REGULARITY EFFECT WITH A DIFFERENT ARCHITECTURE

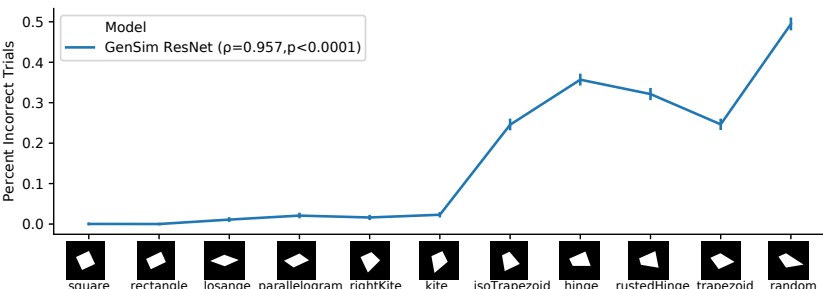

Figure S1: To show the regularity effect can generalize to a different architecture, we finetuned ResNet-101 (He et al., 2016) on the GenSim objective and show that it recovers the human geometric regularity effect. Note that in their supplement, Sablé-Meyer et al. (2021) also reported that a pretrained ResNet-101 fails to produce the geometric regularity effect.

## M  NO GEOMETRIC REGULARITY EFFECT FOR STANDARD CONTRASTIVE LEARNING

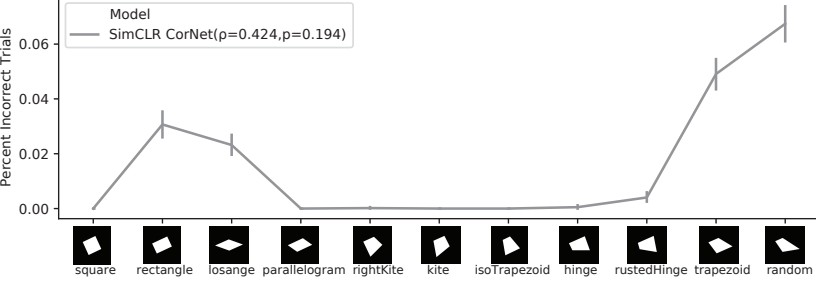

Figure S2: We finetune CorNet using the standard contrastive objective in SimCLR (Chen et al., 2020). Specifically, simple augmentations (cropping and resizing, rotations, etc) were applied to individual quadrilateral images, and then the CNN was trained to push its representations of those images together, to be more similar (i.e., less distant) to their augmented counterparts, and pull its representations of different quadrilateral images apart, to be more dissimilar (i.e., more distant) from each other. Finetuning on this objective does not result in the human geometric regularity effect. Note that the performance of this network is much higher than that of other models and both humans and baboons. To understand why, recall that the reference image choices in the Oddball task are different scales/rotations of the reference image. Because the SimCLR objective applies similar image augmentations to minimize the distance of an image's embedding with its augmented counterpart, the overlap between the training paradigm and the Oddball task allows the network to overfit to the Oddball task. Crucially, however, this network is not *human-like* because it lacks the regularity effect and therefore does not have the human inductive bias we strive to instill in this work.

# N    CONSISTENCY OF GEOMETRIC REGULARITY EFFECT ACROSS DIFFERENT SUBJECTS

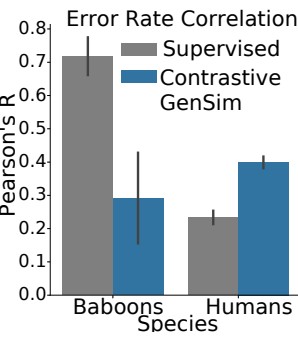

Figure S3: **Consistency of Figure 3C results across different subjects** Correlation between mean finetuned models' error rates with individual human or monkey error rates. This is a reproduction of Fig. 3C, but with behavior correlations for individual subjects instead of the average over subjects. Error bars denote 95% confidence intervals over different subjects (different monkeys or different humans).

# O    REPRODUCING DRAWING STYLE EXPERIMENTS WITH A DIFFERENT ARCHITECTURE

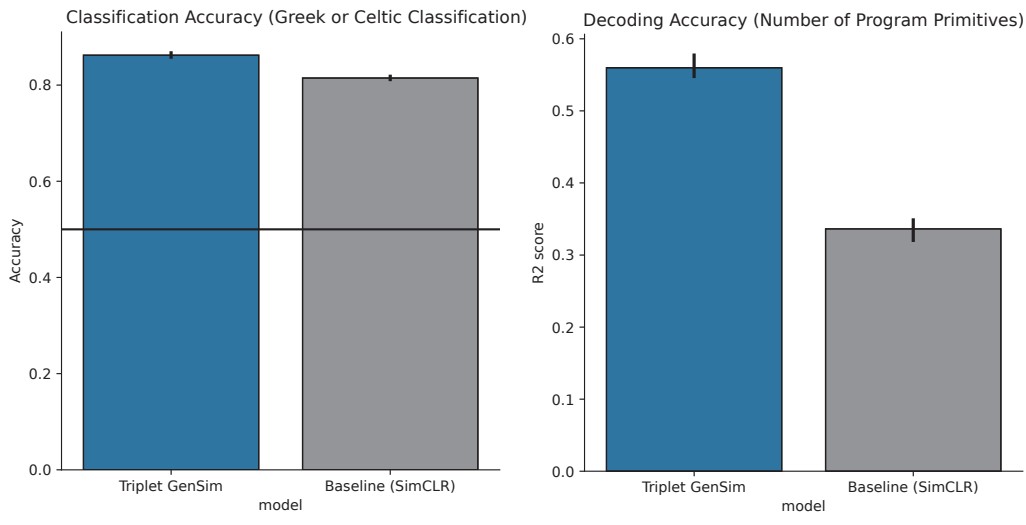

Figure S4: To show the generality of the results in Figure 4, we reproduced the results with a different CNN architecture (CorNet) than the one reported in Figure 4. With a different architecture, training on the GenSim contrastive objective still can decode Greek or Celtic drawing style better (left) and predict the number of motor and control primitives used (right). Errorbars are 95% confidence intervals over different model training runs.

## P   HIERARCHICAL CATEGORIES EXPERIMENT FOR TRAINING SET CATEGORIES

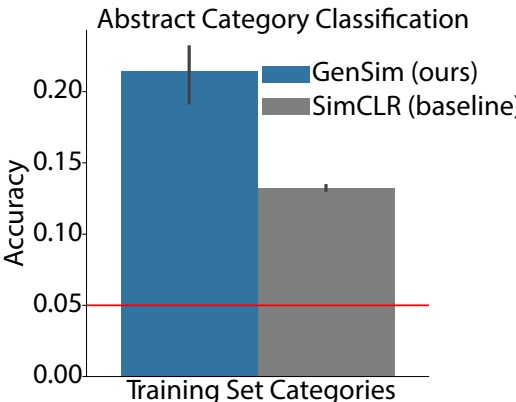

Figure S5: For the result in Figure 6, we show the performance for GenSim-trained networks on predicting image categories that were in the training distribution. Errorbars are 95% confidence intervals over different model training runs. Red line indicates chance levels. Note chance is lower on predicting training categories than test, because there were more training categories (20 vs. 8).

## Q   REPRODUCING HIERARCHICAL CATEGORIES EXPERIMENT WITH A DIFFERENT ARCHITECTURE

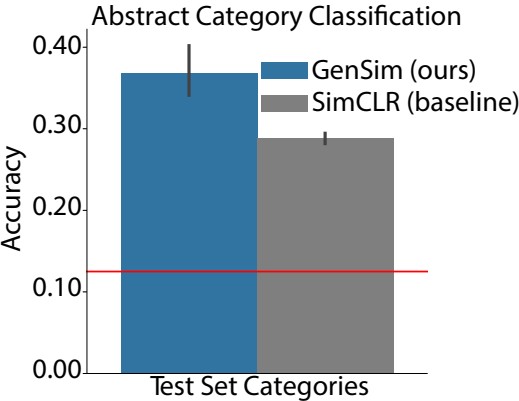

Figure S6: To show the generality of the results in Figure 6, we reproduced the results with a different CNN architecture (CorNet) than the one reported in Figure 6. With a different architecture, training on the GenSim contrastive objective still can decode high-level categories that were unseen during training. Errorbars are 95% confidence intervals over different model training runs. Red line indicates chance levels.

