# OpenReview forum: "Using Contrastive Learning with Generative Similarity to Learn Spaces that Capture Human Inductive Biases"
_ICLR.cc/2025/Conference — ICLR 2025 Conference Withdrawn Submission_

### Official Review · Reviewer_sniL · 2024-10-29

**Soundness:** 1
**Presentation:** 2
**Contribution:** 2
**Rating:** 3
**Confidence:** 3

**Summary:**

The paper proposes a supervised metric learning strategy to enhance representation learning. Specifically, assuming access to a ground truth hierarchical generative model, the authors outline a strategy to sample positive and negative pairs, which are then utilized in a contrastive objective to train the neural network. Experiments are conducted across various scenarios, including synthetic experiments to discriminate modes in Gaussian mixtures and a simple psychological test to distinguish between geometric shapes, demonstrating the possibility of computing the objective directly without relying on sampling-based estimation. Further experiments focus on differentiating style and underlying program rules from handwritten drawings and categorizing abstract concepts in tieredImageNet. The proposed solution is compared with a self-supervised contrastive baseline (SimCLR) to show that supervised information introduces stronger inductive biases than the unsupervised data augmentation strategies in SimCLR, leading to improved predictive accuracy in downstream tasks.

**Strengths:**

- - The paper is well-written and clear **Clarity**
- The problem of instilling human inductive biases through supervision is relevant and timely **Relevance**
- A wide range of experimental tasks are conducted to outline the benefits of the proposed metric learning strategy over vanilla self-supervised contrastive learning **Significance**
- Code is provided with the submission, however no further check has been performed **Code availability**

**Weaknesses:**

- The first major weakness is in the methodology and the underlying message of the paper **Quality**. The paper focusses on demonstrating that data augmentations in contrastive learning are not sufficient to capture human inductive biases compared to direct supervision. This is demonstrated by comparing the proposed solution against a self-supervised contrastive learning strategy, SimCLR. However, such comparison is not fair for two reasons: 1. It is unfair to have a supervised vs. unsupervised comparison. The authors should provide the same amount of supervision to the baseline in order to demonstrate the benefits of their proposed objective. 2. The authors should provide a comparison with existing supervised deep metric learning strategies, as ultimately this is the problem considered in the paper.
- It is not clear what is the underlying novelty of the paper compared to deep metric learning, as discussion and comparison is missing **Novelty/Quality**. At a more conceptual level, the novelty is rather limited and incremental. The proposed objective is essentially a traditional metric learning objective, as it boils down to have the following form $E_{x,x^+,x^-\sim p(X,X^+)p(X^-)}\{\ell(x,x^+,x^-)\}$. Is the novelty related to the way supervision is provided? The benefits should be discussed and supported through experiments.
- Important related work is missing **Quality**. First of all, discussion about the relation with deep metric learning should be provided. Moreover, there are recent works on the formal definition of generative similarity and Bayesian interpretations of self-supervised learning, see for instance [1,2]
- The experimental methodology lacks essential details, which undermines its soundness **Quality/Soundness**. Very limited information is provided about training, raising doubts about the general validity of the results, especially for the comparison with the baseline. For instance, on tieredImageNet which ResNet architecture has been used and why is training performed only on 10 epochs? This is not common practice in self-supervised learning. Authors should test their solution on traditional self-supervised learning benchmarks like ImageNet-1k to better support their claims against contrastive learning.

**References** \
[1] Kim, Puthawala, Chul Ye, Sansone. (Deep) Generative Geodesics. In ICML 2024 Geometry-grounded Representation Learning and Generative Modeling Workshop \
[2] Sansone, Manhaeve. GEDI: GEnerative and DIscriminative Training for Self-Supervised Learning. arXiv 2022

**Questions:**

Please, refer to the above weaknesses.

---

### Official Review · Reviewer_JdTJ · 2024-11-02

**Soundness:** 3
**Presentation:** 3
**Contribution:** 2
**Rating:** 6
**Confidence:** 3

**Summary:**

The paper proposes a new contrastive learning objective based on Bayesian notions of generative similarity.
The authors evaluate the effectiveness of this new objective on a variety of applications, from synthetic to more naturalistic.
This brain-inspired objective outperforms other similar baselines such as SimCLR.

**Strengths:**

-  The paper is clear and well written, and very easy to read for the readers.
- The contributions are strongly backed up with elegant theoretical arguments.
- The authors propose a variety of tasks and models on which this new objective can effectively be applied -- highlighting the flexibility and efficacy of their method.

**Weaknesses:**

- Sec 3.2, 3.3, and 3.4, the generative similarity objective aligns a lot with the goal of the tasks for the model. Optimizing for it surely can be considered an inductive bias in the models, but it is consequently not surprising to observe that the models outperform the baseline without this objective, and even similar objectives such as SimCLR (as the similarity in SimCLR is object-wise instead of category based).
Have the authors considered investigating some emergent properties due to this objective, such: as training on the most regular shapes and testing on the least regular in Sec. 3.2; testing on another style of drawing in Sec. 3.3; or evaluating the robustness of the model in Sec. 3.4. These suggestions are examples of ways to evaluate whether this objective brings any new properties to the models outside of the scope of the loss function. They do not need to be necessarily tested by these types of experiments would strengthen the claims of the authors.

- The authors mention that this objective can be considered as an additional step towards better alignment with humans. But the link with humans is very tangential in this article, except in Sec 3.2. To validate the role of such an objective in the human brain, have the authors considered specific types of behavioral studies or neural datasets they could use to further validate the human-like properties of the learned representations?

**Questions:**

- Instead of SimCLR, other contrastive methods define positive samples as samples belonging to the same category and negative samples from different categories [1,2]. Is it possible to include one of them as a baseline?

[1] Khosla, Prannay, et al. "Supervised contrastive learning." Advances in neural information processing systems 33 (2020): 18661-18673.
[2] Sohn, Kihyuk. "Improved deep metric learning with multi-class n-pair loss objective." Advances in neural information processing systems 29 (2016).

- Sec 3.4, do the 8 testing categories overlap with the 20 training categories? And what about the classes?

- Also, why are there two levels of abstraction (top and bottom levels)? What happens with 100% of top or 100% of bottom?

---

### Official Review · Reviewer_oraS · 2024-11-03

**Soundness:** 2
**Presentation:** 3
**Contribution:** 1
**Rating:** 3
**Confidence:** 3

**Summary:**

The current work proposes a form of contrastive learning based on generative modeling that can target different information than typical contrastive learning based on augmentations. Specifically, if one assumes a discrete number of generating distributions, “generative similarity” is taken to be the odds ratio that two samples come from the same distribution versus different ones.

In the experiments, generative similarity is primarily used as motivation for the manner of producing positive and negative samples for use with standard triplet losses.  Four case studies are investigated: 1) a simple two-Gaussian generative distribution, 2) a study into reproducing human-like visual processing of simple shapes, 3) line drawings produced by a probabilistic program trained on Greek and Celtic, and 4) hierarchical labels on an ImageNet analogue.

**Strengths:**

The paper is well written and the experiments cover a decent range, with a focus on reproducibility.  The experiment on reproducing a quirk of human visual processing (expt 2) is interesting.

**Weaknesses:**

The experiments offer little support that the central innovation---generative similarity---has practical value.  Standard triplet losses are used for all but the experiment about trying to replicate aspects of human visual processing (2).  Experiments 3 and 4 involve relevant representation learning tasks, but they compare contrastive learning where triplets are relevant to the task at hand to triplets formed via standard image augmentations (SimCLR).  It’s not surprising that the former is more successful: if you are able to group the data into positives and negatives that relate to the task you want to solve, better performance isn’t coming from a helpful inductive bias, it’s coming from supervision.

Tying similarity to a specific generative model makes the method’s practical utility highly dubious, and the experiments do not shed light on how accurate the assumed generative model must be for the similarity to yield useful representations.  If generative similarity is only useful when one already has a perfect generative model for the data, I don’t see why it’d provide additional value.

**Questions:**

Can the authors please clarify how to view incorporating knowledge of a dataset-specific generative model as instilling an inductive bias, rather than leveraging supervision?

How can the tiered imagenet example be seen as leveraging an inductive bias and not "human similarity judgements [that] are expensive to scale" (L18)?

---

### Official Review · Reviewer_7CMg · 2024-11-03

**Soundness:** 2
**Presentation:** 3
**Contribution:** 2
**Rating:** 5
**Confidence:** 4

**Summary:**

This paper introduces a novel method for incorporating human-like inductive biases into machine learning models by integrating Bayesian models of cognition within a contrastive learning framework. By defining similarity based on the probability that two data points are generated from the same distribution, a concept called “generative similarity”, the authors aim to overcome the computational challenges of traditional Bayesian approaches and the scalability issues of incorporating human judgments directly. This method is empirically tested across four domains: Gaussian mixtures, quadrilateral shapes, probabilistic programs, and ImageNet.

**Strengths:**

**Originality and Significance.** The paper aims to solve an interesting and important problem of inducing human inductive biases in neural networks. This is especially important for developing more general and intelligent systems.

**Quality.** The paper is very well written and multiple experiments are conducted to support their approach.

**Clarity.** The paper is well-organized and clear, with understandable figures. Key insights and background information are provided wherever appropriate.

**Weaknesses:**

- For the experiment on the tieredImageNet dataset, the authors assume access to abstract categories for each image. However, the comparison is done with SimCLR with random augmentations, where no such information is utilized. SimCLR uses positive samples as random augmentations of the anchor image since it doesn’t have access to image labels or secondary information that can help with the sampling procedure. It might be more fair to compare with say SupCon (Supervised Contrastive Learning) which uses ground truth image labels to improve the representation quality.

  Moreover, for the bottom level of abstraction when sampling triplets, the authors construct the positive image as another sample from the same category and same class as the anchor, and the negative image as sampled from the same category but independent class. This bears similarities to hard negative sampling in contrastive learning (https://arxiv.org/abs/2010.04592) which again improves the representation. A more fair comparison should be made with such baselines, assuming access to the same amount of secondary information.

  Furthermore, the authors merely present the accuracy of both approaches for Abstract Category classification. How does generative similarity affect the quality of representation in terms of performance on downstream tasks (linear probe), nearest neighbors (KNN), etc? This analysis is specifically important to show how human-based inductive biases help in learning a richer representation.

- While there are interesting experiments demonstrating proof of concept and the key idea, there is no comparison with existing baselines (such as https://proceedings.neurips.cc/paper_files/paper/2023/file/9febda1c8344cc5f2d51713964864e93-Paper-Conference.pdf and many more). It is unclear what is the quantitative impact of this learning paradigm and how it compares theoretically or empirically to these existing models.

- I find it very interesting how the authors are able to propose for instilling inductive biases in neural networks without the need to collect human similarity judgments at scale. However, their approach assumes access to a hierarchical generative model of the data, which is a significant and potentially limiting assumption. Obtaining such data at scale can be just as resource-intensive, and in many cases, characterizing a hierarchical generative model is challenging.

**Questions:**

See the Weaknesses section above

---

### Official Review · Reviewer_TbHL · 2024-11-04

**Soundness:** 2
**Presentation:** 2
**Contribution:** 2
**Rating:** 3
**Confidence:** 3

**Summary:**

This paper proposes an approach to instilling human-like inductive biases in machine learning models by combining generative similarity with contrastive learning. While the idea is interesting, questions remain about its scalability and applicability to more complex, real-world settings. The method is primarily tested in simplified tasks, which makes it hard to determine its real-world implications.

**Strengths:**

- The idea of integrating a Bayesian generative similarity measure into contrastive learning is new.
- The experiments include various tasks, such as geometric reasoning, probabilistic programming, and large-scale image classification, showing the method’s flexibility.

**Weaknesses:**

- The framework relies on manually provided inductive biases. This raises concerns about scalability and feasibility for more complex or real-world applications.
- The method is confined to the triplet loss only, which limits its usability for other settings.
- I feel the usefulness of the method is not well shown overall. The paper only demonstrates that the network trained on generative similarity performs better than just a single baseline of SimCLR, in a simple setting. If the performance is the goal, I think there are a lot of ways to improve it for the given task. Additionally, while the advantages of instilling inductive biases into machine learning models, such as improved accuracy, few-shot learning, interpretability, and robustness, were mentioned in the intro, only performance gain in very simple settings is shown in the paper.

**Questions:**

- How does the proposed method scale up to more complex real-world problems?
- Can you show the advantages mentioned in the intro, such as few-shot learning, interpretability, and robustness, through the experiments performed other than improved performance in simple settings?

---

### Note · Authors · 2024-11-27

**Comment:**

We thank the reviewers for their time and evaluations. After careful consideration, we decided to withdraw our submission and we will take the reviewers’ comments into account in future iterations of the present work. We would like to make a few comments however:

1. Multiple reviewers brought up the lower amount of baselines. Our larger point in this work is a novel theoretical re-characterization of contrastive learning as learning embeddings based on a probabilistic generative model, effectively instilling the inductive bias of this generative model into the learned embeddings. Under this characterization, SimCLR corresponds to a default generative process in which a base image is sampled and then a random augmentation is applied. Other CL methods, such as SupCon, would correspond to a generative process that involves sampling from discrete categories and then applying random augmentations. Because of the flexibility of our framework, it can capture many current contrastive learning techniques. In this respect, our goal with the experiments was to show more complex generative processes will lead to stronger inductive biases in the learned embeddings, not necessarily to show we have developed a brand new contrastive learning method that is unique and outperforms all previous techniques and baselines.

2. Some reviewers brought up that our improvements over SimCLR were due to the fact that we imposed more supervision with generative similarity, and were concerned about the difference between instilling an inductive bias vs. providing supervision. We would like to underscore that using supervision is equivalent to instilling an inductive bias from some probabilistic model. In the most extreme case, a regular classifier trained to classify 10 MNIST digits learns the inductive bias of a probabilistic model in which digits come from 10 discrete and independent classes. Moreover, if one defines supervision narrowly as providing a discrete and independent set of labels to specific datapoints, not all supervised schemes guarantee human-like inductive biases. Our framework indeed accommodates a larger variety of ways to provide inductive biases that include but also go beyond supervision (i.e. anything that can be specified by a hierarchical generative model). See our quadrilateral domain as an example where we utilized a hierarchical generative model to produce a network with a more human-like inductive bias than a network trained with vanilla supervision.

3. Some reviewers questioned the value of using contrastive learning with generative similarity when having access to a hierarchical generative model of the data. We respectfully disagree with the reviewers that having the perfect generative model excludes the necessity of neural network models. Here are examples of domains, models, and issues with the models that could benefit from instilling the hierarchical model into a flexible neural network instead: (i) Domain: Language. Model: Probabilistic grammars or other generative models can capture the structure of language with high accuracy. Issue: Parsing long or ambiguous sentences becomes computationally intractable due to the exponential growth of possible parse trees. (ii) Domain: Genomics and Population Genetics. Model: Bayesian models of population evolution (e.g., coalescent models) and genome sequences. Issue: The space of all possible genetic variations and evolutionary histories is vast. Calculating likelihoods of observed data under the generative model often requires simulations that are computationally intensive.

We thank the reviewers again for their time and effort.

**Withdrawal Confirmation:**

I have read and agree with the venue's withdrawal policy on behalf of myself and my co-authors.